# Larval dispersal of Brachyura in one of the largest estuarine/marine systems in the world

**Francielly Alcântara de Lima**[1]*, **Davi Butturi-Gomes**[2¤a], **Marcela Helena das Neves Pantoja**[1], **Jussara Moretto Martinelli-Lemos**[1¤b]

**1** Research Group on Amazon Crustacean Ecology, Federal University of Pará, Belém, Pará, Brazil,
**2** Department of Mathematics and Statistics, Federal University of São João del-Rei, São João del-Rei, Minas Gerais, Brazil

¤a Current address: Federal University of São João del-Rei, Minas Gerais, Brazil
¤b Current address: Nucleus of Aquatic Science and Fisheries of Amazon (NEAP), Federal University of Pará, Belém, Pará, Brazil
* francielly@ufpa.br

**Data Availability Statement:** All relevant data are within the manuscript and its Supporting Information files.

## Abstract

The Amazon Continental Shelf (ACS) is a complex habitat that receives a large annual freshwater discharge into the ocean, producing a superficial plume and carrying with it large amounts of nutrients to the continental shelf along thousands of kilometers while sustaining high biodiversity in the estuary–ocean continuum. For the first time, this study monitored six sites in a wide transect with approximately 240 km radius on the ACS every 2–4 months. The objectives were (1) to analyze the composition of larval Brachyuran crabs and (2) to predict the importance of environmental parameters (temperature, salinity and chlorophyll-*a*) in structuring their abundance. A total of 17,759 larvae identified were distributed in 8 families and 24 taxa. The water salinity was the best predictor of larval distribution. The statistical models used indicated that Panopeidae and Portunidae larvae are more frequent and more likely to occur in shallow water layers, while Calappidae occur in deeper layers, and Grapsidae, Ocypodidae, Sesarmidae, Pinnotheridae and Leucosiidae occur similarly in both strata. The larval dispersal extent varies among families and throughout the year while the groups are distributed in different salinities along the platform. The probability of occurrence of Portunidae is higher in ocean water ($\geq$ 33.5); Grapsidae, Panopeidae, and Pinnotheridae is higher in intermediate and ocean salinity waters (25.5 to 33.5); Ocypodidae, Sesarmidae and Calappidae is higher in estuarine and intermediate salinity waters (5 to 25.5), whereas Leucosiidae, euryhaline, occur in all salinities (5 to 33.5). Furthermore, the Amazon River seasonal flow and plume movement throughout the year not only regulate the larval distribution and dispersion of estuarine species but are also fundamental for the ACS species, providing the necessary nutrient input for larval development in the region.

## Introduction

The various aspects of the Brachyura larval cycle, such as morphology, tolerance to environmental factors, and geographic distribution have been studied from the 1960s to the present

**Funding:** We are grateful to the Brazilian National Research Council, CNPq, for supporting this project (INCT AMBIENTES MARINHOS TROPICAIS Edital MCT / CNPq / FNDCT n° 71/2010). We would like to thank the PROPESP/FADESP (PAPQ Program) for financing the translation of the original manuscript by the native fluency English speaker Ruth Kakogiannos. We thank the students of the Amazonian Crustacean Ecology Research Group (GPECA) and other colleagues for assisting field sampling and sample processing, especially Prof. Dr. Fernando Abrunhosa for the confirmation of some larval groups. The funders had no role in study design, data collection and analysis, decision to publish, or prepara-tion of the manuscript.

**Competing interests:** We confirm: a. That the document is original and all authors agree to its submission and the Corresponding author (Francielly) has been authorized by co-authors; b. This Article has not been published before and is not concurrently being considered for publication elsewhere; c. The importance of the study to crustacean biology and estuaries, especially on Amazon; d. This article does not violate any copyright or other personal proprietary right of any person or entity and it contains no abusive, defamatory, obscene or fraudulent statements, nor any other statements that are unlawful in any way; e. There are no conflicts of interest.

day [1–3]. The larval dispersal strategies (retention and export) of estuarine crab species are known to be strongly correlated with vertical migration in the water column [4, 5] since the speed and direction of horizontal currents vary with depth [6, 7]. The changing vertical position of larvae, either by daily or ontogenetic migration, allows using the stratified currents for displacement in different distances and directions [8], the so-called Selective Tidal Stream Transport (STST) [9, 10].

Dispersal strategies have been identified especially from laboratory experiments used to assess larva tolerance to varying abiotic factors and based on these results suggest the adopted migratory behavior. This type of study has already been carried out for crabs of the families Ocypodidae [11], Panopeidae [12], Varunidae [13], among several others. However, obtaining a successful experiment simulating environmental conditions and guaranteeing complete larval development, especially counting on obtaining ovigerous females whose eggs are in hatching conditions is not an easy task, which certainly contributes to the lack of data for most species. On the other hand, investigating larval distribution in a natural environment (estuary and continental shelf) and providing valuable information on the species life cycle and migratory behavior, also presents several difficulties, especially in countries where research investment is low, justifying the low number of publications on this topic compared to studies developed with the adult population in the same environments [14–20].

This scenario is aggravated in the vast and complex Amazon region, which harbours one of the largest aquatic diversities on the planet so that a significant portion of species is still unidentified and has unknown distribution limits [21–23], except for some relatively well documented commercial fish groups [24–26] and, recently, zooplankton [18], shrimps [27] and thalassinoids [28]. In estuarine and coastal Amazonian regions, data on the distribution of Brachyura larvae and their regulatory factors are extremely limited [29] and practically nonexistent for the extended continental shelf ($\approx$ 240 km from the coast) [30], one of the widest in the world. This gap can be explained, mainly, by the high financial cost of sampling and strong currents, which hinder access to the area and limit the knowledge on the Brachyura fauna life history and geographical distribution.

This megadiverse and conspicuous environment of the Amazon Continental Shelf (ACS) [26] drains an average discharge of 5.7 x $10^{12}$ m$^3$/year from the Amazonas/Araguaia/Tocantins hydrographic system [31]. This freshwater mass from the continent forms a surface plume on the continental shelf, whose trajectory changes throughout the year due to the interaction of different factors, such as the Amazon River discharge, North Brazil Currents (NBC), North Equatorial Countercurrent (NECC) and Guyana Current (GC), the atmospheric Intertropical Convergence Zone (ITCZ), as well as winds and tides [32, 33]. In addition to the enormous water discharge, a large supply of organic matter and sediment, approximately 900 to 1150 x $10^6$ ton/year [34, 35], greatly affects the planktonic dynamics and biomass of the adjacent coastal and Atlantic regions [36, 37], making the Amazon estuary unique among estuaries worldwide [38].

To date, the occurrence of 194 species of adult Brachyura has been recorded in the ACS, of which only 74 have some larval stage described [39] while 34 species in their benthic phase are constantly captured as accompanying fauna by the industrial fishing fleet with emphasis to the pink shrimp *Farfantepenaeus subtilis* (Pérez-Farfante, 1967) [40]. In this study, the larval period of Brachyura was evaluated regarding (1) parameters of larval composition/distribution and (2) how these relate to environmental profiles. The first parameters were analyzed by the larval dispersal extent related to the estuary/plume, and by the frequency of occurrence (FO) in the plume categories. The latter was predicted by the probability of occurrence (PO) and abundance for the temperature, salinity, and chlorophyll-*a* profiles in the ACS with a multimodel approach.

Our objective is to improve the knowledge on biodiversity of this important component of the food web, as well as to characterize the group configuration in the plankton of the continental shelf, revealing possible larval dispersal strategies, distribution, and life cycle in the largest estuarine-marine region of the world. It was expected that all larval developmental stages of each species should be distributed on their parental populations over the ACS.

## Material and methods

### Study area

The investigation and the field studies did not involve endangered or protected species. All applicable international, national, and/or institutional guidelines for the care and use of animals were followed. The study was conducted on the Amazon Continental Shelf (ACS), specifically in the area affected by the plume of the Amazon and Tocantins/Araguaia Basins. We sampled in six different locations along the coastal area of Marajó Island to near the slope, from 23 to 233 km away from the coast (Fig 1), and ranging from 2–4 months from July 2013 to January 2015. The Amazon River discharge to the ACS has strong seasonality, with approximately 220,000 and 100,000 $m^3 s^{-1}$ maximum and minimum flows in May and November, respectively [41, 42]. Additionally, the Amazon plume that remains close to the continental margin in the winter in the Northern Hemisphere (January to March / April), begins to spread to the north during the peak discharge of the Amazon River in the spring (May). In the summer (June to July), the NBC retroflection takes the plume to the east and, in the fall (September), 70% of the plume water is exported to the east via this route [33, 43].

### Data collection

Two sets of standardized samples were collected during daytime and syzygy from 6 sampling stations differing in their distance from the coast and two different sampling depths (horizontal, oblique), but also different seasons (7 expeditions in 2 years). Two zooplankton hauls were conducted: one horizontal subsurface at 0.5 m from the surface and another oblique ('V'-shaped, covering up to 75% of the local depth, approximately 10, 19, 34, 39, 53 and 80m depths off the coast to the slope). Horizontal and oblique hauls in each site at different coastal distances (23, 53, 83, 158, 198, and 233 km), were chosen because they represent regions under the greatest influence of the Amazon estuarine plume (23–83 km from the coast) and also to contemplate the large extension of the Continental Shelf (approximately 300 km) with a distance of 158–233 km with a predominance of ocean waters. Both hauls used a plankton trawl consisting of a 2 m long conical net with 200 μm mesh, 60 cm opening diameter, and a coupled flow meter. The hauls lasted five minutes at a speed of approximately 2 knots ($\approx$ 4 km / h). Before plankton samples, the environmental variables water temperature (˚C), salinity, and chlorophyll-*a* (μg / L) contents were obtained using a CTD probe (Hydrolab DS 5) that was used to get samples every 0.5 meters.

The obtained 84 samples (7 expeditions x 6 sites x 2 hauling methods), with an initial volume of 500 mL each, were preserved in formaldehyde buffered with sodium tetraborate, 4% final solution. In the laboratory, the samples were fractionated into 250 mL aliquots with a Folsom-type subsample. All brachyuran larvae in the aliquots were dissected and identified to the lowest taxonomic level possible [1, 45–61] by observing the morphological parameters, such as the arrangement and number of spines and setae in the antennae, antennules, maxilliped, maxilla, abdomen, and telson, under an Axioscope Zeiss A1 optical microscope (Carl Zeiss, Oberkochen, Germany). The recent taxonomy was checked in the World Register of Marine Species [http://www.marinespecies.org]. After identification, the non-dissected larvae were

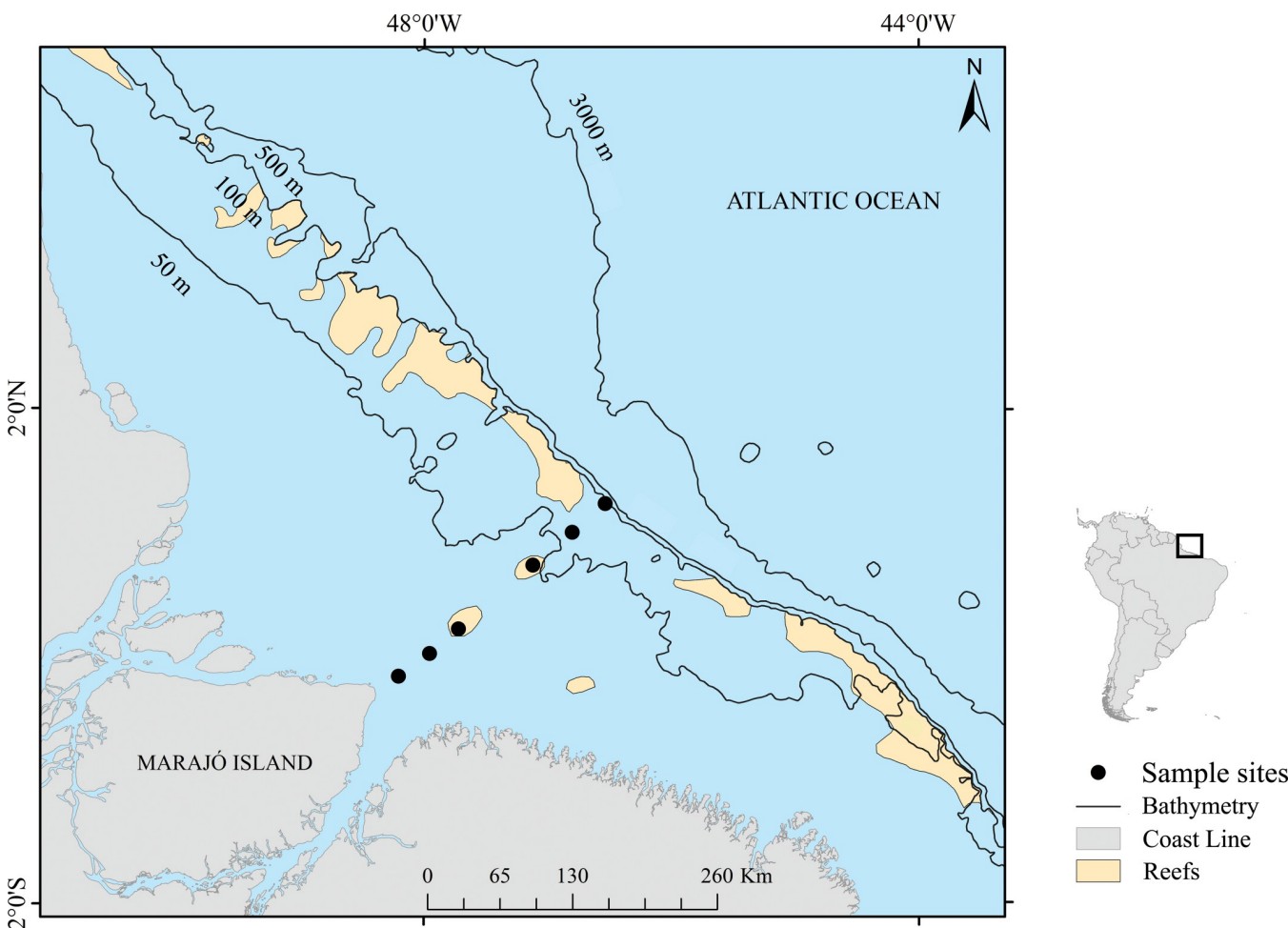

**Fig 1. The Amazon continental shelf, northern Brazil, and the six sampling sites surveyed (23 km, 53 km, 83 km, 158 km, 198 km, and 233 km offshore, from the NE coast of Marajó Island).** Reef shapefile [44]. Reprinted from [Moura et al., 2016] under a CC BY license, with permission from [Nils Edwin Asp Neto], original copyright [2016].

deposited and registered in the Carcinological Collection of the Museu Paraense Emílio Goeldi (MPEG).

## Data analyses

Density was estimated (larvae m$^{-3}$) by dividing brachyuran larval abundance by the volume filtered through the plankton net. The volume was given by the number of rotations of the Hydrobios flowmeter attached to the net aperture, based on the difference between the initial and final numbers observed in each trawl, and calculated as V = A × R × C, where A = net aperture area (A = π.r$^2$), R = number of flowmeter rotations during the haul (Df–Di, final and initial numbers, respectively), and C = measurement factor (m rotations$^{-1}$ = 0.3) determined after device calibration.

The frequency of larval occurrence (FO %) was given for each species as FO = a × 100/A, where a = number of samples containing the species, and A = the total number of samples. The results were categorized as follows: very frequent (FO ≥ 70%), frequent (30 ≤ FO < 70%), infrequent (10 < FO < 30%), and sporadic (FO ≤ 10%) [62].

The expected abundance of each species group (either family or genus, depending on the lowest taxonomic level identified) was predicted by an intensive multimodel approach [63]. We fit several different predictors to each dataset, corresponding to each group abundance data. Given the presence of many zeros in our datasets–some of which might simply be structural zeros–we conditionally predicted the expected abundance in the predicted probability of occurrence of each species group using the presence-absence data. The details of each step of the data analysis are given below.

To the presence-absence data, we fit binomial generalized linear models with logistic link-function [64] using different predictors. We computed the predictors by taking the powerset (excluding, of course, the empty set; please check [65] for clarifications on this issue) of the available covariables (plankton net depth, temperature, chlorophyll-*a*, distance from the estuary, salinity, and their squared values) and used each element of the powerset to fit the models, which yields 511 different models. For each model, we computed the Akaike weights:

$$\omega_j = \frac{exp\{-\Delta_j/2\}}{\sum_{i=1}^{M} exp\{-\Delta_i/2\}},\tag{1}$$

where $\omega_j$ is the Akaike weight for the $j$-th model and $\Delta_j = AIC_j - min_{i \in M}\{AIC_i\}$ is the scaled AIC for the $j$-model in a set of $M$ models. The value of $M$ is the difference between the number of elements in the powerset of covariables (511) and the number of models that achieved convergence; thus, $M$ may have varied greatly between any two species groups. Finally, the probability of occurrence (PO) was predicted based on all models weighted by their corresponding Akaike weights. We used, as a reference scenario for the predictions, the median values of temperature and chlorophyll-*a* and three different values of salinity (5, 23, and 33.5) for estuary distance varying between 23 and 233 km (respectively, the nearest and the farthest sites).

Given the predicted PO using the presence-absence data, we fit Poisson generalized linear models with log-link function to the observed abundance data using the previous rationale (several models, each one with one element of the powerset of covariables, weighted by the Akaike weights, discarding models that did not converge, computing the predictions on the reference scenario). Since the observed abundances largely depend on the water volume collected in the sample, we set the volume as an offset in all models and used the median for the predictions.

The final prediction for the expected abundance of each group is given by the product between its predicted abundance and the PO in the reference scenario. It is noteworthy that our multimodel and information-theoretical approach does not allow (and it would be meaningless) to report p-values of any kind [63, 68], thus we used the total weight of each variable across all plausible models, varying from 0 (never important) to 1 (always important), as a measure of influence on the predictions. For clarity, we considered a model to be plausible if its $\Delta_j < 2$, which, within the information-theoretical context, means that there is strong empirical evidence that the model is a good approximation to reality (low loss of information) compared to the pool of all models under consideration (see [63, 66–68], for instance).

## Results

### Environmental variables

The water temperature range in the ACS was 7˚C (23 to 30˚C), with the highest value recorded in January (beginning of the more intense rain period) and the lowest in July (beginning of the less rainy period). The lowest temperatures were recorded at great depths, from 200 km away from the continent. However, a temperature of about 28˚C was predominant in most of the

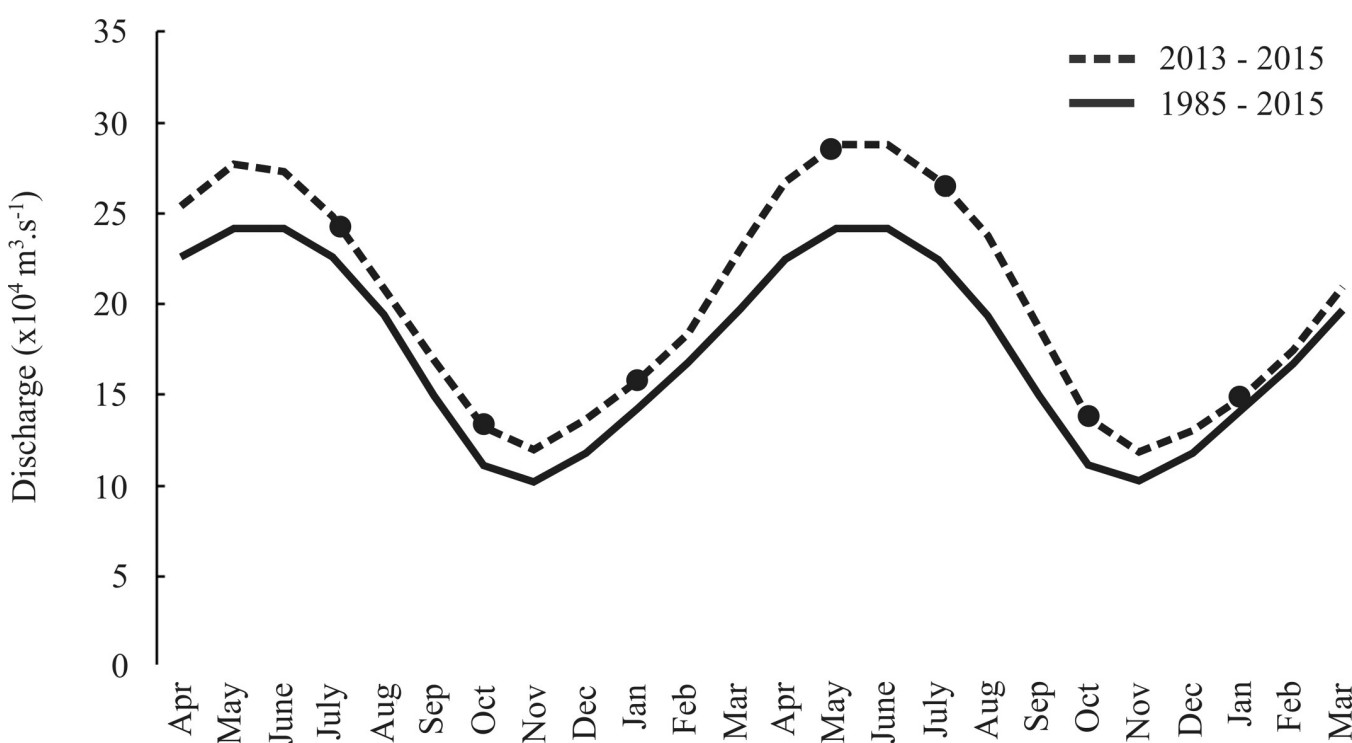

**Fig 2. Historical outflow variation of the Amazon River (Óbidos station, Pará) from 1985 to 2015 provided by the Brazilian national water agency (ANA) (http://www.snirh.gov.br/hidroweb/).** Black circles indicate field sampling.

shelf extension. The water salinity varied widely (range of 36), with a minimum of 2 (May, rainy season) and a maximum of 38 (January), while increasing gradually away from the coast. The chlorophyll-*a* content ranged from 0.3 to 85.1 μg/L$^{-1}$, in January, decreasing as the coast-ocean distance increased (for detailed environmental variables in this region, see [27]). The Amazon River outflow was higher (291,900 m$^3$.s$^{-1}$) in May and lower ($\approx$110,000 m$^3$.s$^{-1}$) in October 2013–2014, with a water volume variation similar to the historical average from 1985 to 2015 (Fig 2, S1 Table).

## Larval species/family composition

A total of 17,759 identified larvae were distributed in eight families and 24 taxa (Table 1).

*Panopeus lacustris* Desbonne in Desbonne and Schramm, 1867 was the most abundant species (67% FO, of which 57% megalopa), followed by *Achelous* spp. De Hann, 1833 (12%) and *Armases rubripes* Rathbun, 1897 (9%). Both phases of larval development (zoea and megalopa) of *P. lacustris*, *A. rubripes*, Gelasiminae Miers, 1886 and Portunidae Rafinesque, 1815 were found in the ACS. Also, the occasional occurrence of *Leptuca cumulanta* (Crane, 1943) ZI and ZII stages, and *Ucides cordatus* (Linnaeus, 1763) and *Goniopsis cruentata* (Latreille, 1803) ZI were also observed but not included in the mathematical models, whereas the other species larval stages occurred heterogeneously on the ACS (Table 1). *Achelous* spp., *A. rubripes*, *Calappa* sp. Weber,1795, *Callinectes* spp. Stimpson, 1860, Gelasiminae 2, *Pachygrapsus gracilis* (de Saussure, 1857), *P. lacustris*, and *Pinnixa* sp. White, 1846 larvae occurred over the entire study period. Whereas *Hexapanopeus* spp. Rathbun, 1898 zoea did not occur in May only and Portunidae n. id. did not occur during the lower outflow in October. Additionally, *Minuca rapax*

**Table 1. Larval composition and frequency of occurrence of Brachyura on the Amazon Continental Shelf.** Legend according to [69] that characterize the Amazon River plume depending on salinity: E = estuarine plume (0 < Salinity ≤ 20), IP = intermediate plume (20 < Salinity ≤ 31), OP = outer plume (31 < Salinity ≤ 36), OO = open ocean (Salinity > 36). The frequency of occurrence increases as follows white ≤ 10%, light gray ≥ 10–30%, dark gray ≥ 30–70%, and black ≥ 70%.

| Taxon | Stages | Month | Hauls | Plume | | | |
|---|---|---|---|---|---|---|---|
| | | | | E | IP | OP | OO |
| **Calappidae** | | | | | | | |
| *Calappa* sp. | ZI–ZIV | Oct/14 | O | dark gray | | light gray | black |
| **Grapsidae** | | | | | | | |
| *Goniopsis cruentata* | ZI | Oct/13 | O | | black | | |
| Grapsidae n. id. | ZII–ZIV | May/14 | O | | | | black |
| *Pachygrapsus gracilis* | ZI–ZIII | Oct/13 | O | light gray | | | dark gray |
| **Leucosiidae** | | | | | | | |
| Leucosiidae n. id. | ZII–ZIV | Jul/14 | SS | | | light gray | black |
| *Persephona* spp. | ZI–ZIV | Oct/13 | O | light gray | | | black |
| **Ocypodidae** | | | | | | | |
| Gelasiminae 1 | ZI | May/14 | SS | dark gray | | dark gray | |
| Gelasiminae 2 | ZI | May/14 | SS | dark gray | light gray | light gray | light gray |
| Gelasiminae 3 | ZI | May/14 | SS | dark gray | dark gray | dark gray | |
| Gelasiminae n. id. | ZII–ZVI | Jan/14 | SS | light gray | dark gray | dark gray | light gray |
| *Leptuca cumulanta* | ZI, ZII | Oct/13 | O | | black | | |
| Megalopa 1 | | Jan/14 | O | dark gray | dark gray | | light gray |
| Megalopa 2 | | Jul/14 | SS | black | | | |
| *Minuca rapax* | ZI, ZIII, ZIV | May/14 | SS | dark gray | | | |
| *Uca maracoani* | ZI, ZIII, ZIV | Jul/13 | O | | | black | light gray |
| *Ucides cordatus* | ZI | Jan/15 | SS | black | | | |
| **Panopeidae** | | | | | | | |
| *Hexapanopeus* spp. | ZI–ZIV | Jan/14 | SS | | light gray | light gray | dark gray |
| Megalopa | | Jan/15 | SS | | | | black |
| *Panopeus lacustris* | ZI–ZIV, M | Oct/13 | SS | light gray | | light gray | black |
| *Panopeus* sp. | ZI–ZIV | Oct/13 | SS | | | light gray | black |
| **Pinnotheridae** | | | | | | | |
| *Austinixa* sp. | ZI–ZV | Jan/14 | SS | | | dark gray | black |
| *Dissodactylus crinitichelis* | ZI–ZIV | Jan/15 | SS | | | light gray | black |
| *Pinnixa* sp. | ZI–ZV | Jan/14 | SS | light gray | | dark gray | black |
| Megalopa | | Jan/15 | O | | | | black |
| **Portunidae** | | | | | | | |
| *Achelous* spp. | ZI–ZVII | Jul/13 | SS | | | | black |
| *Callinectes* spp. | ZI–ZVIII | Jul/13 | SS | light gray | | | black |
| Megalopa 1 | | Jul/14 | SS | light gray | | light gray | black |
| Megalopa 2 | | Jan/14 | O | | | | black |
| Portunidae n. id. | ZI–ZIII | May/14 | SS | light gray | | light gray | dark gray |
| **Sesarmidae** | | | | | | | |
| *Armases rubripes* | ZI–ZIV, M | Jan/14 | SS | dark gray | light gray | light gray | light gray |

ZI = zoea I; ZII = zoea II; ZIII = zoea III; ZIV = zoea IV; ZV = zoea V; ZVI = zoea VI; ZVII = zoea VII; ZVIII = zoea VIII; M = megalopae; n. id. = not identified; SS = sub-superficial; O = oblique.

(Smith, 1870), Gelasiminae 1, 2 and 3 larval density peaked during the higher river outflow in May (see S1–S8 Figs).

### Larval dispersal (depth strata, coastal distance, and estuarine plume)

The mathematical model results indicate different patterns of larval dispersal along the continental shelf. Regarding the disposition in the water column, Panopeidae Ortmann, 1863, Pinnotheridae De Haan, 1833, and Portunidae were more abundant and had a higher PO in subsurface waters while *A. rubripes* and Ocypodidae larval stages were more abundant in the water column. *Calappa* sp. larvae occurred almost exclusively in the water column, while Grapsidae MacLeay, 1938 and Leucosiidae Samouelle, 1819 abundances were similar in both strata, surface and water column (Figs 3–5).

The occurrence of crabs was heterogeneous, varying according to the distance from the coast and water salinity. Portunidae larvae had a greater abundance and higher PO in waters with 33.5 salinity and furthest from the continent, from 83 km away from the coast (Fig 4D). On the other hand, Ocypodidae (Fig 4E) and *A. rubripes* (Fig 4F) abundance and PO were higher near the coast, up to 158 km, in estuarine and intermediate plume waters, with 5 and 25.5 salinity, respectively. Grapsidae (Fig 3A), Panopeidae (Fig 3B), and Pinnotheridae (Fig 3C) larval stages were also more abundant in coastal waters, however, with higher PO in waters with 25.5 and 33.5 salinity. *Calappa* sp. larvae occurred with especially higher density at 158 km (Fig 5G) between 83 and 233 km from the coast, mainly in waters with 5 and 25.5 salinity, while Leucosiidae larval distribution (Fig 5H) was similar throughout the continental shelf therefore, occurring in several salinities.

During the highest flow period, *Pinnixa* sp., Grapsidae, Panopeidae and Portunidae larvae were concentrated in the mid and outer continental shelf since the freshwater plume "pushes" the larvae to locations away from the coast. The opposite was observed during the lowest outflow, these larvae concentrated near coastal areas (23 and 53 km from the coast) due to the lower freshwater outflow entering the ACS (Figs 6 and 7, S2 and S3 Tables). Furthermore, few taxa were not affected by the Amazon plume. Leucosiidae had the highest larval density on the middle continental shelf (158 km) while the density of Ocypodidae was also the highest up to 158 km in the highest flow (May) and decreased/reduced in the lowest flow period (October). The *A. rubripes* larvae occurred particularly in three sites closest to the coast (23, 53, and 83 km) with few variations in the period and among months whereas *Calappa* sp. larvae occurred 158 km away from the coast, near reef areas, regardless of varying river flow, and *D. criniticnelis* Moreira, 1901, occurred only between 83 and 158 km away from the coast (S2 Table).

## Discussion

The results indicate that the larval dispersal of Brachyura along the ACS is structured by the Amazon River plume, whose outflow reaches the highest and lowest volume in May and October, respectively. Larval species that hatch near the coast are subject to the same abiotic factors as light, temperature, pressure, and salinity [70], and biotic factors such as predators and food availability [71, 72]. However, each taxon can adjust its vertical position up or down the water column according to the changing exogenous stimuli [10, 73, 74], affecting the direction and the distance that the larvae are horizontally transported [19, 75]. These different responses are probably related to characteristics intrinsic to each species or family, such as adult habitat [76], osmoregulation [77], presence of predators [78], among other factors.

In the export process, brachyuran larvae are carried along the plume edge towards the ACS. In the maximum flow conditions, the Amazon plume can extend up to 300 km from the coast [79] and up to 10 meters deep [80], with salinity below 35 [81].

The pattern observed in other world regions suggests that temperature is one of the most important seasonal factors controlling the biological processes within plankton populations and communities [82, 83]. In this study, however, temperature and chlorophyll-*a* were not

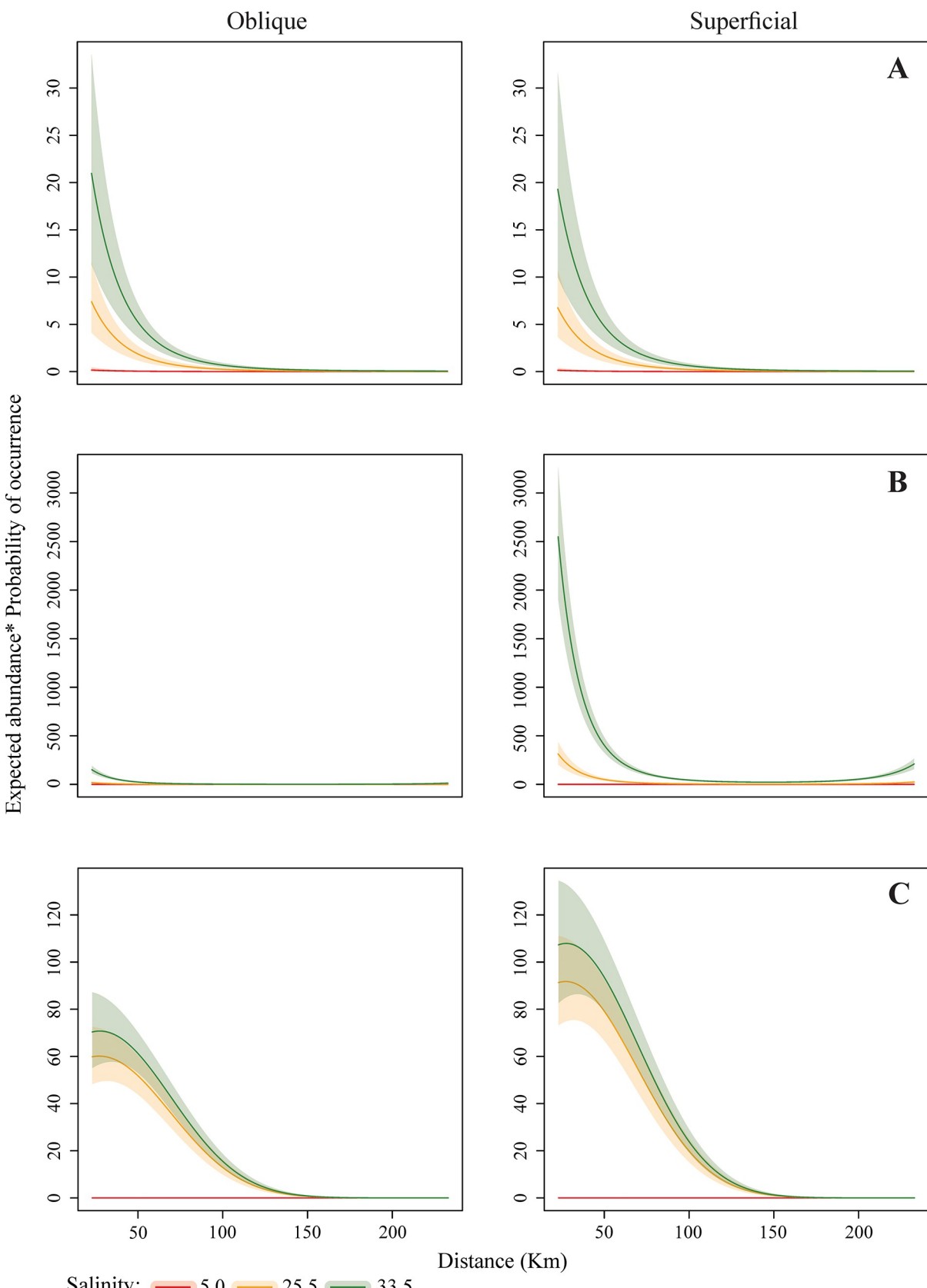

**Fig 3.** Final expected abundance (product between the expected abundance and the probability of occurrence) of Brachyura larvae (A: Grapsidae; B: Panopeidae; C: Pinnotheridae) according to the distance from the estuary and net depth in different salinities. The shaded regions refer to the weighted 95% confidence interval. All predictions used the observed median values of temperature (28.4°C), chlorophyll-*a* (7.34 μg / L) and water volume (278.3 m$^{-3}$).

important dispersal predictors so the different larval strategies seem to be mainly regulated by the strong influence of the Amazon River outflow on the water salinity and the Amazon plume expansion and retraction throughout the year (S4 Table). Increases in meroplankton abundance are likely to be ascribed to high nutrient input and primary production, e.g. chlorophyll-*a* between 4 and 5 (mg/m$^3$) [15] and the fact that chlorophyll-*a* is widely available in PCA, this factor does not explain the variation in larval abundance. Although plume flow was not closely followed in this study, its influence on the salinity is notorious and, consequently, on the larval community of brachyurans on the continental shelf, diverging from other coastal regions due to the peculiar characteristic and strength of this hydrological system. Other factors frequently related to the movement of planktonic larvae, such as acting tides, winds, and surface currents [19, 84] were not addressed in this work, but these important parameters should be investigated in future research. In this context, the larval dispersal of each family and its distribution in the water column on the ACS is detailed below.

## Grapsidae

The Grapsidae zoea (*P. gracilis* and Grapsidae n. id.) is distributed similarly in the water column and surface layer, with higher expected abundance and PO in waters with salinity between 25.5 and 33.5, estimated to be more coastal. However, the larvae moved away from the coastal region, occurring only in 198 and 233 km during the highest discharge of the Amazon River (May) but moved closer to the continent, where water salinity is higher compared to the plume, during the lowest flow period.

Our results confirm the larval export proposed for *P. gracilis* [85], however, PO is higher in more coastal areas, a behavior also adopted by others *Pachygrapsus* Randall, 1840 [5, 86, 87], *Hemigrapsus* Dana, 1851 [88], and *Geograpsus* Stimpson, 1858 [89]. Similar to *P. gracilis*, larvae of the *Pachygrapsus* and *Hemigrapsus* genera also go through all the larval developmental stages in the mid-continental shelf [90, 91]. Some zoeas like *P. crassipes* zoeae are distributed in all water column strata, surface, and bottom [70], whereas others such as *Hemigrapsus* zoeae have negative geotaxis and swim towards the surface [75], spending the entire larval cycle about 15m deep [92], revealing that the vertical arrangement varies among different genera and species of grapsids in the water column.

The *P. gracilis* zoea I, II and III were the most abundant larvae of the family in the ACS, and occurred every month, confirming the continuous reproduction, similar to *P. transversus* [93, 94]. However, larval peaks for the species were observed on the ACS and estuary in October and May, respectively [29, 85]. *P. gracilis* zoeae seem to have a high tolerance to varying salinity since three of at least five zoea stages that make up the genus [95, 96] are found over the entire ACS extension (23 to 233 km) and in different salinity strata. No megalopa of this species was found in the ACS, despite the information that they develop on the continental shelf [97] and immigration is correlated with high salinity and current speed, typical of estuary flood tides. Future work should clarify such gaps regarding other grapsids in ACS.

## Panopeidae

Panopeidae larvae are distributed mainly in the surface layer of the water column, both zoea and megalopa had a higher predicted abundance and PO associated with intermediate and

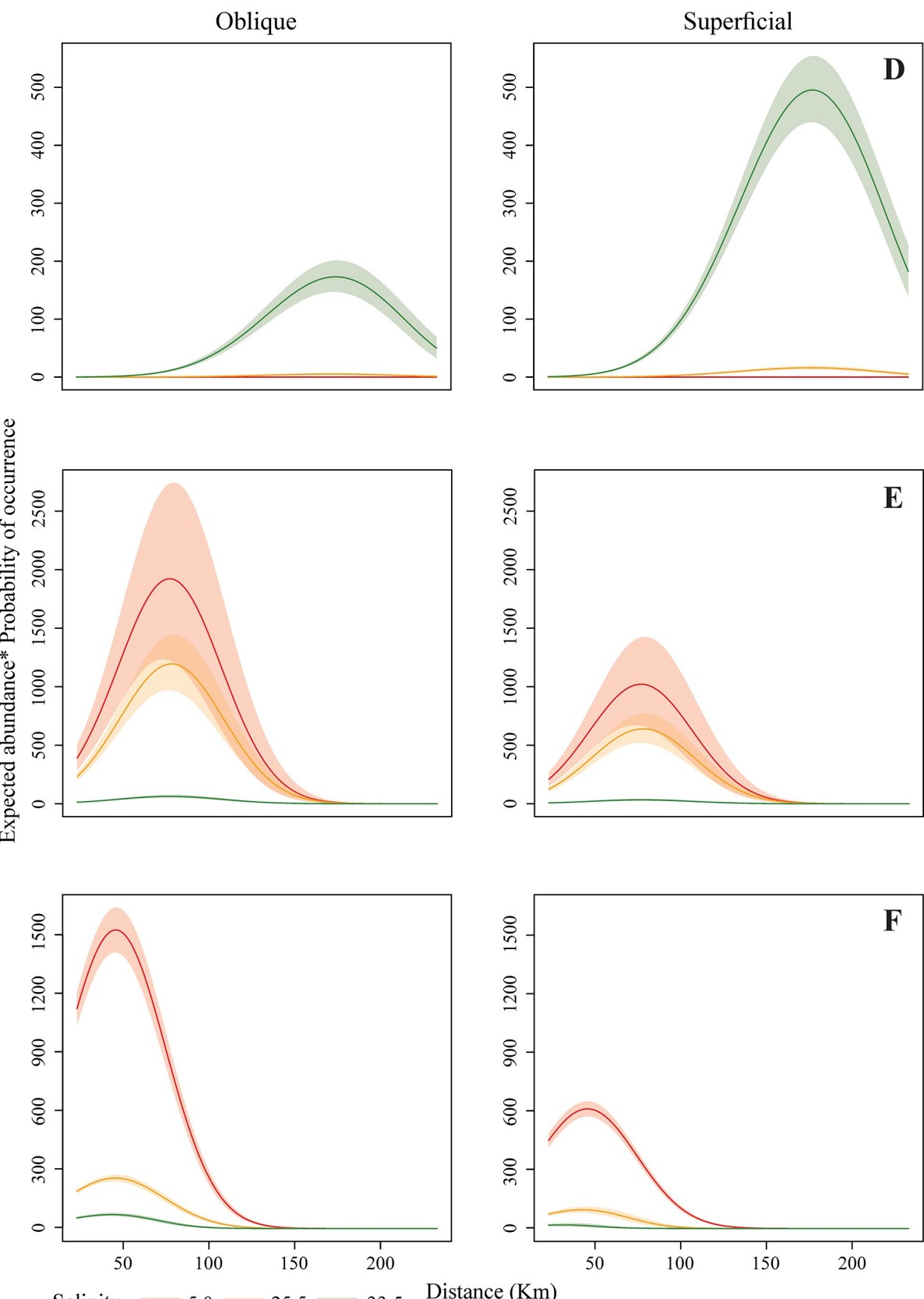

**Fig 4.** Final expected abundance (product between the expected abundance and the probability of occurrence) of Brachyura larvae (D: Portunidae; E: Ocypodidae; F: Sesarmidae) according to the distance from the estuary and net depth in different salinities. The shaded regions refer to the weighted 95% confidence interval. All predictions used the observed median values of temperature (28.4°C), chlorophyll-*a* (7.34 μg / L) and water volume (278.3 m$^{-3}$).

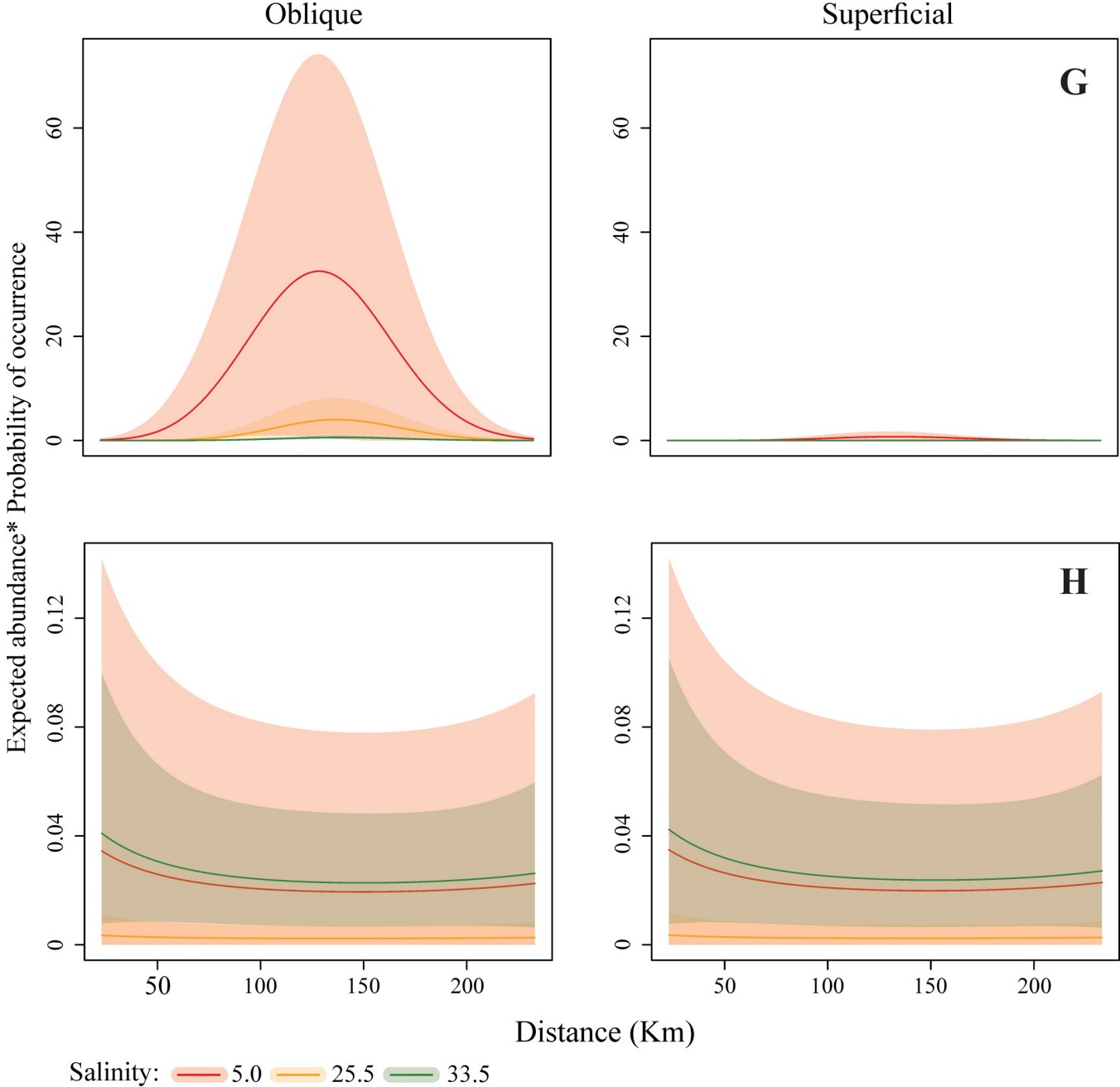

**Fig 5.** Final expected abundance (product between the expected abundance and the probability of occurrence) of Brachyura larvae (G: Calappidae; H: Leucosiidae) according to the distance from the estuary and net depth in different salinities. The shaded regions refer to the weighted 95% confidence interval. All predictions used the observed median values of temperature (28.4°C), chlorophyll-*a* (7.34 μg / L) and water volume (278.3 m$^{-3}$).

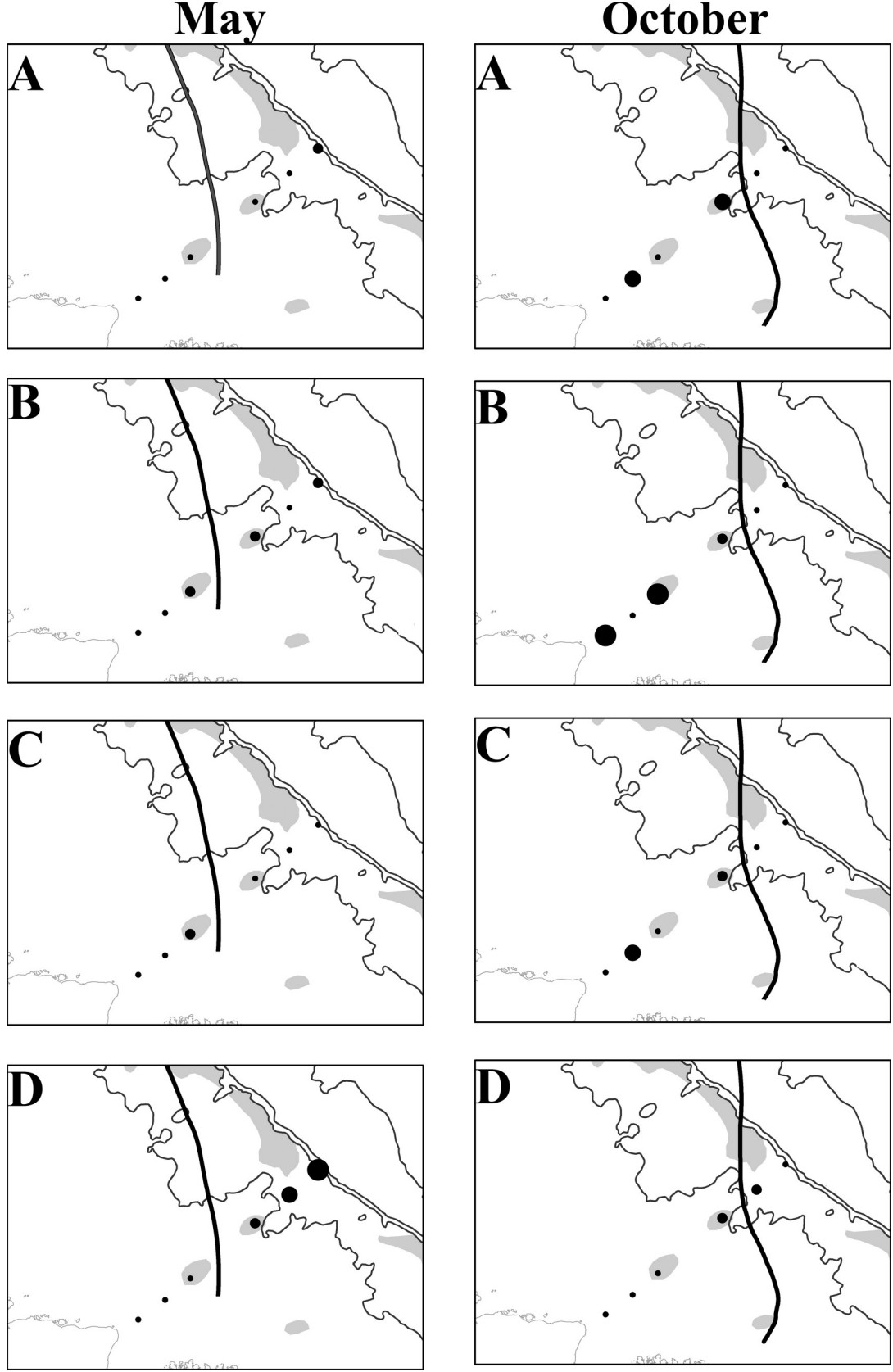

**Fig 6. Density of Brachyura larvae on the ACS in May and October 2014.** Grapsidae (A), Panopeidae (B), Pinnotheridae (C), and Portunidae (D). Bold line: plume-ocean ecotone (44). Reprinted from [Moura et al., 2016] under a CC BY license, with permission from [Nils Edwin Asp Neto], original copyright [2016].

oceanic salinities (25.5 and 33.5), as well as lower and higher density in May and October, respectively. Because *Panopeus* H. Milne Edwards, 1834 larvae occur in the full extent of the ACS (≈ 240 km), they seem to have greater tolerance to ocean waters compared to *Hexapanopeus* spp. [54]. The latter also performs larval export, but dispersal extends up to 158 km from the coast, with all its zoea stages (I to IV) developing on the mid-continental shelf.

## Pinnotheridae

The predicted abundance and PO for Pinnotheridae larvae were slightly higher in surface waters and with intermediate and oceanic salinity (25.5, 33.5) while all zoeal stages were present in the ACS. Further, *Pinnixa* and *Austinixa* Heard and R.B. Manning, 1997 larval dispersal reached half the extension of the continental shelf (158 km), and were present throughout the year, with a density peak in January, consistent with the reproductive habits already reported for other species [98–101].

Pinnotheridae larval export has already been suggested based on the high abundance of zoea I larvae in the superficial layer during the estuary low tides, in addition to a positive correlation between larval density and salinity [102], which is in line with our results. In California (USA) estuaries, this migration of zoea I to the continental shelf was observed for *Pinnixa faba* (Dana, 1851), *P. tubicola* Holmes,1895, *Pinnotheres pugettensis* Holmes, 1900, and *Scleroplax granulata* Rathbun, 1894, where later stages zoea developed at a depth of 17 m while megalopae, between 49 m and 6 km from the coast [92]. In the Chesapeake Bay, all larval stages of *Pinnixa chaetopterana* Stimpson, 1860, *Pinnixa sayana* Stimpson, 1860, *Tumidotheres maculatus* (Say, 1818) and *Zaops ostreum* (Say, 1817) were always found close to the bottom, indicating larval retention [2, 103]. Likewise, larval retention for *Pinnixa gracilipes* was also suggested in an Amazonian estuary due to the presence of ZI, ZII and ZIV in the lower estuary [29].

In our study, only the *Pinnixa* sp. distribution was influenced by the Amazon River outflow since larvae occurred 83 km and 23 km from the coast during high and low discharge, respectively. The *Austinixa* sp. distribution did not vary much while *D. crinitichelis* zoeae were found in October only, 83 and 158 km from the coast, places that overlap the coral reefs of the ACS, potential habitats of several Echinoidea, host organisms of the adult crab (mainly the Orders Clypeasteroida and Spatangoidea) [56].

In this context, we believe that the different patterns may be related to the host habitat, since Pinnotheridae consists basically of symbiotic crabs or parasites of benthic invertebrates, which inhabit thalassinoids burrows [98], Polychaeta tubes [99], bivalves [104], echinoderms, among others. And because the symbiotic relationships can vary from quite host-specific to very non-specific [105], there will likely be species adaptations to the different host environments, which should be further investigated in more detail.

## Portunidae

In the ACS, we found all larval stages of *Achelous* spp. and *Callinectes* spp., as well as Portunidae larvae distributed in the superficial layer, with greater abundance and PO from 83 km from the coast and in high salinity (33.5), where the later developmental stages are highly abundant. Also, larvae of this family are found more distant (83 to 233 km) and closer (23 to 233 km) to the coast during the high and low outflow of the Amazon River, respectively,

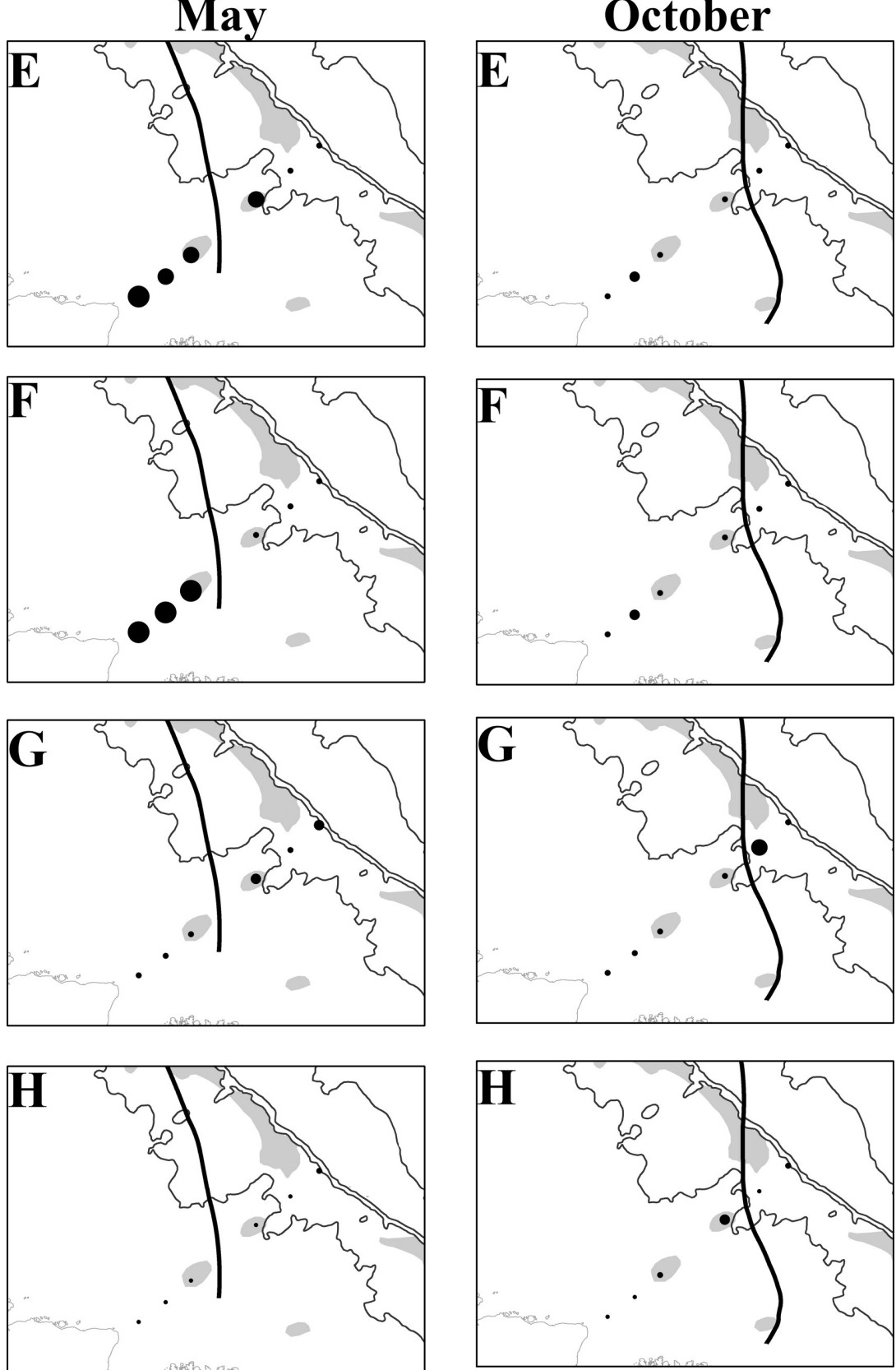

**Fig 7. Density of Brachyura larvae on the ACS in May and October 2014.** Ocypodidae (E), Sesarmidae (F), Calappidae (G), and Leucosiidae (H). Bold line: plume-ocean ecotone (44). Reprinted from [Moura et al., 2016] under a CC BY license, with permission from [Nils Edwin Asp Neto], original copyright [2016].

indicating their affinity with ocean waters while confirming this group previous work on the larval stages.

We assume that export strategy [106], is probably followed by the *Callinectes* larvae that leave the Amazon estuaries close to the plume toward the continental shelf. Whereas marine species larvae that inhabit the ACS, such as *Achelous* and *Cronius*, for example, normally develop in the ocean salinity range [3], from hatching to settling in the benthic or free-swimming habitat of adults. In both cases, inhabiting high salinity waters is essential for the successful larval development of these species [107, 108].

The larvae identified here as *Achelous* spp. are believed to correspond mostly to *Achelous rufiremus* (Holthuis, 1959), which is the most abundant benthic macroinvertebrate species at almost four degrees of latitude in the ACS and is the most abundant in the bycatch pink shrimp fishing [40, 109]. However, the description of the larval morphology is not available to confirm this hypothesis. This problem can be solved in the future by using DNA barcoding analysis to identify larvae originating from the natural environment.

## Ocypodidae

In the ACS, all Ocypodidae larval stages were present, with zoea and megalopa concentrated up to 158 km and 83 km away from the continent, respectively. The highest predicted abundance and PO were recorded in estuarine and intermediate salinity waters (5 and 25.5), with a greater dispersal extent (up to 233 km) and reproductive peak during the Amazon River high outflow (May), as observed in the Amazon estuary [29].

This family export strategy to the adjacent continental shelf is already known [4, 110–112]. Except for *L. cumulanta*, which retains all larval stages in the estuary [29], so that ZI and ZII larvae may have been accidentally dragged by currents to the continental shelf, thus justifying the single occurrence 23 km from the coast.

The Ocypodidae larval density was higher in deep compared to superficial waters. Similar to observations for *Minuca pugnax* (Smith, 1870) larvae [11] and other ocypodids, where ZI and the later zoea stages occur close to the surface and deeper in the water column, respectively, suggesting an ontogenetic alteration in the vertical position [103], allowing these larvae to take advantage of the gravitational circulation for eventual transport of the megalopa back to the estuary [16], and to settle juveniles and adults in the same habitats [113, 114].

The *U. cordatus* species is a peculiar case because has an inconspicuous cycle, which breeds seasonally in the Amazon, specifically in the rainy season full or new moon [115, 116]. Our sample covered two reproductive cycles of this crab, despite that, only three ZI specimens were found at the estuary mouth (23 km) in January. This low abundance probably results from the species high synchronization associated with the action of currents, since *U. cordatus* always spawns at ebb tide in the night, in new or full moons [117], and its synchronization is so strong that one day later upon release, ZI larvae are no longer found in the estuary [116]. Thus, once on the continental shelf, the larval transport may have been enhanced since, in January, the land winds blow the Amazon plume against the continental margin, which follows quickly with the North Brazil Current towards the French Guiana [33, 80], justifying not being captured by our nets.

Therefore, unlike *U. cordatus*, the possibility of finding Gelasiminae larvae is always greater since its several spawnings throughout the year guarantee a constant larval supply to the continental shelf.

## Sesarmidae

This semi-terrestrial crab *A. rubripes* has continuous reproduction with a peak of ovigerous females in the summer (December, January, and February) [118]. Unlike adults living in oligohaline estuaries, larvae are not as tolerant to low salinity so that full development is only possible in more saline waters, suggesting migration to the continental shelf [119, 120]. Our results confirm larval export, with all larval stages dispersed up to 123 km from the coast, and reproductive peak in January, the high rainfall period in the Amazon region. The rainfall intensity seems to be the parameter regulating *A. rubripes* reproduction in the equatorial region, as observed for *Aratus pisonii* (H. Milne Edwards, 1837) [121, 122] and *Armases angustipes* (Dana, 1852) [123], which can be advantageous to these populations due to increased concentration of nutrients, higher tides—increasing the chance of transport over long distances—and increased seawater productivity, which favors the larval development in plankton [121, 124].

In the ACS, the abundance and PO predicted for *A. rubripes* zoea and megalopa are higher in deeper water, mainly in estuarine salinity (5), with lower occurrence in intermediate salinity (25.5). These salinity values are consistent with those used in the species cultivation for the larval description (about 19) [60] but differ from the high survival range of zoea (30) for specimens grown in Uruguay [120]. Such disagreement seems to indicate that the saline tolerance varies among larval populations from different locations, adapting to the low salinity resulting from the great discharge of freshwater from the estuaries while the plume seasonality and river flow do not affect their distribution in the plankton in the Amazon region. Besides, the megalopa was observed concentrated up to 83 km away from the coast, seemingly opting for even lower salinity waters compared to the zoea, a pattern previously observed in megalopae of *A. pisonii* [125] and *Armases* spp. [126, 127], and also followed by other crab species that remigrate from the continental shelf to estuaries and rivers.

## Calappidae

The zoea of *Calappa* sp. occurred on the platform throughout the year, indicating continuous reproduction. The high predicted abundance and PO in deeper layers of the water column indicate that zoeae were practically absent from the surface layer. This genus inhabits coral, sand and mud bottoms, and is exclusive to the continental shelf [39]. The larvae were concentrated mainly in the coral reef regions (83 to 158 km from the coast), while adults [40] were distributed over the entire reef system area, estimated between 9,500 and 56,000 km$^2$ [44, 128]. Thus, these environments, fundamental to the development of *Calappa*, are greatly affected by humans since 1960 due to predatory bottom trawling fishing to capture the pink shrimp, which destroys the reefs and captures the genus as bycatch [40, 109], and the future possibility of oil and gas exploration in the area [128].

Although adults inhabit only the continental shelf, and the genus larval distribution is not affected by the seasonality of the river flow, we find that *Calappa* zoea seems to have some affinity with plume water, perhaps by the injection of land-derived sediments, nutrients, and organic matter dissolved in the ocean environment [129]. This transition region, where the plume front meets the ocean water is an ecotone (83 to 158 km), intricately linked to the river discharge flow, being one of the ACS areas with the greatest amplitude of primary productivity when the concentration of nutrients and irradiance conditions are optimal, especially in the

maximum flow period [43, 81], to support larval marine species, such as *Calappa*, living close to the transition zone.

In the ACS, four species of the genus are found in the adult population [40]: *Calappa nitida* Holthuis, 1958, *Calappa ocellata* Holthuis, 1958, *Calappa sulcata* Rathbun, 1898, and *Calappa gallus* (Herbst, 1803). The latter is the only species in the region with the ZI stage described [45], which limited the identification of the specimens.

In our study, we found zoeae with morphological characteristics corresponding to the ZIV, based on this, we believe that the larval development of the *Calappa* species in the ACS consists of ZI to IV and megalopa, which should be confirmed in later larval descriptions.

## Leucosiidae

The expected abundance and PO for Leucosiidae zoea were high in waters with different salinities (5, 25.5, and 33.5) while not differing regarding the vertical arrangement in the water column. This response may be related to the combination of distinct characteristics of the two groups found. The *Persephona* Leach, 1817 zoeae were distributed more homogeneously along with the ACS and not only close to the coastal region (23 km) whereas Leucosiidae zoeae were present only away from the coast (from 83 km). Despite this, during the largest river flow, the occurrence of both taxa was restricted to 233 km from the coast, and their distribution was more homogeneous in October and January.

Similar to *Calappa*, this group inhabits the sandy and mud bottoms, and coral reefs on the continental shelf, it occurs almost all year round and is also caught as accompanying fauna in shrimp fisheries [40]. Of the eight Leucosiidae genera inhabiting the ACS, only the larval development of *Persephona mediterranea* (Herbst, 1794) has been completely described (ZI–ZIV and megalopa) [48] while *Persephona lichtensteinii* Leach, 1817 and *Persephona punctata* (Linnaeus, 1758) has been only the ZI described in the literature [49]. Little is known about the developmental stages of the life cycle of these crabs, so this is the first information about the larval ecology of the group in a natural environment.

## Amazon specificity

The process of zoeal larvae leaving from estuaries and returning later as megalopa has been extensively studied and reviewed worldwide [9, 10, 106, 130]. The larval supply of Brachyura is continuous, with few exceptions as *U. cordatus*, because the ACS is located in a tropical Amazon region, where summer lasts throughout the year, unlike temperate environments with well-defined seasons, where the spring and summer are the periods of highest larval density [130–133].

Despite this, collecting Brachyura larvae in the plankton of this area is still challenging since, besides the large ocean volume in which the larvae can potentially disperse, they are also distributed in larval patches that remain intact from spawning to the megalopa stage [134, 135]. Given this, knowing the larval distribution of families in the different strata of the water column is relevant for future sampling, where oblique hauling has proven to be effective in capturing Ocypodidae larvae, *Armases* and *Calappa*, while surface hauling is ideal for sampling Panopeidae, Pinnotheridae and Portunidae in the ACS.

The Amazon fluvial discharge affects the dispersion of estuarine families while the Amazon plume influences the species of the continental shelf, being responsible for the continuous distribution of nutrients on the ACS [79] and increasing the primary ocean production in the area. Moreover, the area connecting the ACS and the coast consists of mangroves of the Amazon River mouth, considered a Ramsar site, is extremely important for numerous aquatic organisms, whether as habitat, feeding zone, or nursery [136, 137].

## Conclusion

Our study provides the first documentation on the larval dispersal of Brachyura on the Amazon Continental Shelf. Additionally, it is the first time that data on expected abundance is predicted for each group and presented as the product between the predicted abundance and the probability of occurrence. Crab larvae showed different distribution and dispersal boundaries, adapting to this particular environment. The group of estuarine species *P. gracilis*, Gelasiminae, *U. maracoani* (Latreille, 1802), *P. lacustris*, *Panopeus* sp., *Austinixa* sp., *Pinnixa* sp., *Callinectes* spp. and *A. rubripes* perform larval export to the platform, while *Calappa* sp., Leucosiidae, *Persephona* spp., *D. crinitichelis* and *Achelous* spp., inhabit the platform as adults, especially close to the rhodolith reef region and complete the larval cycle in this same environment. We confirm that the flow of the Amazon River greatly influences the dynamics of the Amazon Continental Shelf, especially the salinity, which is directly related to the larval distribution of Brachyura in this area. We emphasize that further studies on larval description and DNA barcoding could contribute to understanding the Brachyura larval ecology since to protect and value the region biodiversity, the knowledge on species identity, spatial distribution, and correlations with environmental conditions are essential, as these existing gaps create difficulties for identifying and defining priority areas for effective species conservation in the Amazon region.

## Supporting information

**S1 Fig. Density of larval stages of *Calappa* sp., *Goniopsis cruentata*, and Grapsidae n. id. over the months in ACS.**
(TIF)

**S2 Fig. Density of larval stages of *Pachygrapsus gracilis*, Leucosiidae n. id., and *Persephona* spp. over the months in ACS.**
(TIF)

**S3 Fig. Density of larval stages of Gelasiminae 1, 2 and 3 over the months in ACS.**
(TIF)

**S4 Fig. Density of larval stages of Gelasiminae n. id., *Leptuca cumulanta*, and *Minuca rapax* over the months in ACS.**
(TIF)

**S5 Fig. Density of larval stages of *Uca maracoani*, *Ucides cordatus* and *Hexapanopeus* spp. over the months in ACS.**
(TIF)

**S6 Fig. Density of larval stages of *Panopeus lacustris*, *Panopeus* sp., and *Austinixa* sp. over the months in ACS.**
(TIF)

**S7 Fig. Density of larval stages of *Dissodactylus crinitichelis*, *Pinnixa* sp., and *Achelous* spp. over the months in ACS.**
(TIF)

**S8 Fig. Density of larval stages of *Callinectes* spp., Portunidae n. id., and *Armases rubripes* over the months in ACS.**
(TIF)

**S1 Table. Descriptive statistical (minimum, maximum, mean and standard deviation) of environmental parameters (temperature, salinity and chlorophyll-*a*) in relation to months and distance from the coast on the Amazon Continental Shelf.**
(DOCX)

**S2 Table. Larval composition, N total and sum of density (larvae m$^{-3}$, in parenthesis) of Brachyura in each sampling sites on the Amazon Continental Shelf.** "SS" is used as an abbreviation for sub-superficial sample and "O" to oblique hauls.
(DOCX)

**S3 Table. Larval composition, N total and sum of density (larvae m$^{-3}$, in parenthesis) of Brachyura in each expedition on the Amazon Continental Shelf.** "SS" is used as an abbreviation for sub-superficial sample and "O" to oblique hauls.
(DOCX)

**S4 Table. Variable influence on the predicted abundances (sum of the corresponding Akaike weights) across plausible models (AIC difference threshold less or equal to 2).** The values vary from 0 (never important) to 1 (always important). PO: Probability of Occurrence; A: Expected Abundance. FND: Fishing Net Depth; S: Salinity; D: Distance from the coast; C: Chlorophyll-*a*; T: Temperature. The number "two" in their corresponding superscript stands for the variable squared value.
(DOCX)

## Acknowledgments

We are grateful to the native fluency English speaker Ruth Kakogiannos for the translation of the original manuscript. We thank the students of the Amazonian Crustacean Ecology Research Group (GPECA) and other colleagues for assisting field sampling and sample processing, especially Prof. Dr. Fernando Abrunhosa for the confirmation of some larval groups. We are also grateful to the editor and reviewers for comments and suggestions, which substantially improved the manuscript.

## Author Contributions

**Conceptualization:** Francielly Alcântara de Lima, Jussara Moretto Martinelli-Lemos.

**Data curation:** Francielly Alcântara de Lima, Davi Butturi-Gomes, Marcela Helena das Neves Pantoja, Jussara Moretto Martinelli-Lemos.

**Formal analysis:** Francielly Alcântara de Lima, Davi Butturi-Gomes, Marcela Helena das Neves Pantoja, Jussara Moretto Martinelli-Lemos.

**Funding acquisition:** Jussara Moretto Martinelli-Lemos.

**Investigation:** Davi Butturi-Gomes, Jussara Moretto Martinelli-Lemos.

**Methodology:** Francielly Alcântara de Lima, Davi Butturi-Gomes, Marcela Helena das Neves Pantoja.

**Project administration:** Jussara Moretto Martinelli-Lemos.

**Resources:** Jussara Moretto Martinelli-Lemos.

**Software:** Francielly Alcântara de Lima, Davi Butturi-Gomes.

**Supervision:** Jussara Moretto Martinelli-Lemos.

**Validation:** Jussara Moretto Martinelli-Lemos.

**Visualization:** Francielly Alcântara de Lima.

**Writing – original draft:** Francielly Alcântara de Lima, Marcela Helena das Neves Pantoja, Jussara Moretto Martinelli-Lemos.

**Writing – review & editing:** Francielly Alcântara de Lima, Davi Butturi-Gomes, Jussara Moretto Martinelli-Lemos.

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
