## [Decision Letter · Decision Letter 0]

20 Aug 2021

PONE-D-21-16420

Larval dispersal of Brachyura in the largest estuarine / marine system in the world

PLOS ONE

Dear Dr. de Lima,

Thank you for submitting your manuscript to PLOS ONE. After careful consideration, we feel that it has merit but does not fully meet PLOS ONE’s publication criteria as it currently stands. Therefore, we invite you to submit a revised version of the manuscript that addresses the points raised during the review process.

We look forward to receiving your revised manuscript.

Kind regards,

Atsushi Fujimura

Academic Editor

PLOS ONE

Journal Requirements:

We are grateful to the Brazilian National Research Council, CNPq, for supporting this project (INCT AMBIENTES MARINHOS TROPICAIS Edital MCT / CNPq / FNDCT nº 71/2010). We would like to thank the PROPESP/FADESP (PAPQ Program) for financing the translation of the original manuscript by the native fluency English speaker Ruth Kakogiannos. We thank the students of the Amazonian Crustacean Ecology Research Group (GPECA) and other colleagues for assisting field sampling and sample processing, especially Prof. Dr. Fernando Abrunhosa for the confirmation of some larval groups.

6. We note that Figure 1 in your submission contain [map/satellite] images which may be copyrighted. All PLOS content is published under the Creative Commons Attribution License (CC BY 4.0), which means that the manuscript, images, and Supporting Information files will be freely available online, and any third party is permitted to access, download, copy, distribute, and use these materials in any way, even commercially, with proper attribution. For these reasons, we cannot publish previously copyrighted maps or satellite images created using proprietary data, such as Google software (Google Maps, Street View, and Earth). For more information, see our copyright guidelines: http://journals.plos.org/plosone/s/licenses-and-copyright.

7. Please upload a copy of Figure 6 and 7, to which you refer in your text on page 13. If the figure is no longer to be included as part of the submission please remove all reference to it within the text.

Reviewers' comments:

Reviewer's Responses to Questions

**Comments to the Author**

1. Is the manuscript technically sound, and do the data support the conclusions?

Reviewer #1: No

Reviewer #2: Yes

Reviewer #3: Partly

2. Has the statistical analysis been performed appropriately and rigorously? 

Reviewer #1: I Don't Know

Reviewer #2: I Don't Know

Reviewer #3: I Don't Know

3. Have the authors made all data underlying the findings in their manuscript fully available?

Reviewer #1: Yes

Reviewer #2: No

Reviewer #3: No

4. Is the manuscript presented in an intelligible fashion and written in standard English?

Reviewer #1: Yes

Reviewer #2: Yes

Reviewer #3: Yes

5. Review Comments to the Author

Reviewer #1: General comments:

The manuscript deals with the dispersion of Brachyura larvae in an area of the Amazon continental shelf under the influence of the estuarine plume of the Amazon River. Despite being a relevant topic and there is little information available on the subject, the manuscript presents some points that need to be better presented and explained before the manuscript can be published. My opinion is that the manuscript must be rejected in its present form.

Suggestions for Improvements

Major issues

- The manuscript does not have clear objectives and hypotheses.

- The presented methodology leaves several doubts about the adopted procedures.

Pag. 6 line 137. Little information was mentioned about the acquisition of environmental variables, making it impossible to understand how the data were obtained. Data were collected only the surface layer? Other layers of the water column were sampled?

Would the environmental variables obtained in a single layer of the water column be sufficient to understand the complex patterns of larval dispersion on the continental shelf?

Minor issues

Title: “Larval dispersal of Brachyura in the largest estuarine / marine system in the world”

The title of the manuscript does not completely agree with the focus of the manuscript.

Material and methods.

Pag. 4. Line 99. What do the authors consider to be a rare and deep environment?

Pag. 6 line 131. The horizontal samplings were carried out during the daytime, at night time, in which tide? Sampling was standardized?

Pag. 6 line. 132. What are the criteria used in choosing the fixed collection points? How was depth measured?

Pag. 6. Line 134. “V'-shaped, covering up to 75 % of the local depth”. This collection procedure should be better explained.

Pag. 6. Line 137. “Simultaneously, water temperature (ºC), salinity, and chlorophyll-a (μg / L) contents were measured, using a CTD probe (Hydrolab DS 5)”. Environmental variables were obtained from the surface layer? No data were obtained in other depths?

Pag. 6 line 142. What keys were used by the authors to identify the different taxonomic levels? How the diferente stages of larval development were identified?

Results.

In general, the analyzes presented do not clearly demonstrate the relationship of environmental factors in the distribution of larvae.

Pag. 9, line 208. It is not possible to verify how the estuarine plume affected the different sampling points throughout the sampling period.

Discussion.

Some points mentioned above about distribution models may affect manuscript discussion.

Reviewer #2: Please see the attached document for formatted version of reviewer comments below.

This study summarizes field distributions of Brachyuran crab larvae along the Amazon Continental Shelf. The authors describe larval dispersal of diverse taxa across both spatial (depth/distance offshore) and seasonal scales. To date, relatively few studies have reported extensive field distributions of Brachyura during larval development. Hence, this study provides a valuable contribution to better understanding their dispersal patterns.

Please find my comments and suggestions below. Many of my comments are minor suggestions, although there are four major concerns that I hope the authors will address in revision. First, the phrase “larval dispersion” is used incorrectly and should most likely be replaced with “larval dispersal” throughout the manuscript. In ecology, dispersion indicates investigation of specific distribution patterns, e.g. random, clumped, or uniform, whereas “dispersal” indicates the movement of individuals, e.g. export vs. retention. Second, the figures that show model results (Figs. 3 – 5) are missing key information, which make it difficult to confirm the authors’ interpretation of the data. I provided specific comments to this regard below. Third, the paper should include a summary of model outcomes in the results section to support statements made later in the discussion. For example, the authors state that salinity was a reliable predictor of larval dispersal, while temperature and chlorophyll-a were not; however, statistical support for this claim is lacking. Overall, I find it concerning that an extensive multi-model approach was described in the methods section, but there is little description of the outcomes in the results section (aside from Figs. 3 – 5, which require more detail in their respective captions). Fourth, the impact of the seasonal plume is a central point in the paper. However, statistical/graphical support is lacking (see my comment regarding lines 285 – 297 below). The authors reference Figs. 6 and 7 to support this analysis. However, these figures were not included in this submission.

In summary, this study represents a valuable contribution to better understanding the dispersal of larval Brachyurans in coastal systems. However, I recommend that the concerns presented in this review be addressed prior to publication.

Abstract

The importance of the Amazon River seasonal flow and plume is not mentioned until the end of the abstract. Opening the abstract with a sentence or two describing the system would help guide the reader through the results that are summarized just below.

Lines 27 – 31: This sentence runs on a bit. It would be useful to more clearly highlight the two objectives described: (1) to analyze the composition of larval Brachyuran crabs and (2) to predict the importance of environmental parameters in structuring their occurrence/abundance.

Line 31: “A total of 17,759 identified larvae are…” should be “A total of 17,759 identified larvae were…”

Line 39: “(> = 33.5)” should be “(≥ 33.5)”

Line 45: Remove “plankton”

Introduction

Line 73: Do you mean “harbors”, rather than “habirs” here?

Line 81: Why include “among others”?

Lines 99 – 103: I suggest restructuring of this sentence. It might be useful to break it up into (1) parameters of larval composition/distribution and (2) how these relate to environmental profiles. The current organization is a bit convoluted.

Line 104: The “aquatic food chain” is an oversimplification. Consider using “food web” or “trophic interactions” instead.

Line 108: I’m not sure what is meant by “…should be distributed on their parental populations…” Are you expecting close proximity to parental populations? Given the common export strategy of estuarine crabs, would this be likely for all Brachyuran crabs in the region?

Methods

Lines 115 – 116: “July/2013 to January/2015” should be “July 2013 to January 2015”

Line 131: “haulsin” should be “hauls in”

Lines 154 – 155: The density unit should have not a period after larvae, and the sentence could be restructured for clarity, e.g. “Density (larvae m-3) was estimated by dividing Brachyuran larval abundance by the volume filtered through the plankton net.”

Line 161: “…of each species larvae…” should be “…of each larval species…” or “…of each species of larvae”

Line 200: Citation(s) to support that this threshold is “widely used”?

Results

Line 208 and 212: Replace “amplitude” with “range”

Line 209: The parenthetical “begin” should be “beginning” in both occurrences here.

Line 217: Remove the “/” between month and year

Line 248: Add “and” before Pinnixa

Line 285 – 297: I think that the seasonal plume analysis discussed here is important. However, this section requires visual, model, and/or statistical support. Figure 6 and 7 are referenced in this paragraph but these were not included in the submission. In addition, several of the statements made cannot be supported by Table 1 or Figs. 3 – 5, which do not include seasonal information.

Discussion

Lines 301 – 303: To make this statement, more support is needed in the results section (see previous comment regarding lines 285 – 297).

Line 311: Should be “Brachyuran larvae”

Lines 320 – 323: Where are the model results that support this statement?

Line 470: Remove “Anyway”

Line 515: Is “particular” the best word here?

Line 531: Why change the subtitle structure at this point? All others list the family only.

Line 561: Same as above – why alter the subtitle structure?

Line 563: “larval” before zoea is a bit redundant

Line 585: Either add a semicolon after “…(Herbst, 1803)” or start a new sentence, i.e. “The latter is…” The structure of species name reporting also switches to parentheticals here: “Calappa gallus (Herbst, 1803)” rather than the previous “Calappa sulcata Rathbun, 1898”. For consistency, use the same format throughout.

Line 661: “And” should not be capitalized.

Table and Figures

Table 1 only shows the frequency of occurrence for one month/year for each group. Based on the supplemental figures, I assume this is the timepoint with the highest density for each individual taxon. If so, indicate this in the table caption. I also suggest characterizing the colors of the heat map described, i.e. frequency of occurrence increases in order of white, light gray, dark gray, and black. Also, “S” is used as an abbreviation for salinity and for sub-superficial sample. Perhaps, it would be clearer to change the latter to “SS”.

Fig. 1 caption: It would be useful to identify that the distances listed are kilometers offshore, e.g. “…(23 km, 53 km, 83 km, 158 km, 198 km, and 233 km offshore).”

Figs. 3 – 5: More information is needed in the caption, which could also be accomplished by adding a legend. For example, what do the different colors represent – different models? Do the shaded regions around each trendline indicate a confidence interval of some sort? Are the y-axis values shown expected abundance per some unit of volume? The methods state that the “final prediction for the expected abundance of each group is given by the product between its predicted abundance and the PO in the reference scenario”. However, the notation in the y-axis label indicates that you are showing a ratio of predicted abundance / PO, rather than the product.

Supporting information

S9 and S10 were not referenced in the text.

Reviewer #3: Review: "Larval dispersal of Brachyura in the largest estuarine / marine system in the world", submitted to PlosOne by F.A. de Lima et al.

This paper describes seasonal variations and regional distribution patterns in the occurrence of brachyuran crab larvae studied in a transect from 23 to 233 km off the NE coast of Marajó Island, Amazon estuary, Brazil. Each transect comprised 6 sampling stations visited from July 2013 to January 2015 during 7 expeditions in approximately quarterly intervals (actually every 2-4 months). Since the offshore distribution of crab larvae in the Amazon estuary is very little known and considerable amounts of novel data were obtained in this study, this paper should eventually be published in an international journal such as Plos One. Before it can be accepted, however, it should undergo a thorough revision and restructuring. Most of my concerns are related to the description of the methods, the presentation of the data, and the detailed discussion of higher taxa (mostly at the family level). While I feel that some aspects are given too much attention, others are neglected or remain unclear (see below). - I suggest that the authors should carefully consider the following points:

- General organization of this paper: It appears to me (especially in the Abstract and in the Discussion section), that the authors put far too much emphasis on the systematic position of the identified larvae rather than to ecological groups and reproductive strategies. For some of the 25 crab taxa identified in this study, the benthic juvenile and adult life-history stages are well known as to their salinity requirements, living and reproducing either in estuarine (i.e. brackish) coastal habitats or in offshore (marine) waters with higher salinities. The data gathered in this field study, especially those of horizontal and vertical distribution of crab larvae (comparing parallel samples from surface and sub-surface water) should therefore be looked at in more detail, mainly in the context of export and retention strategies (for aims of this study, cf. p. 5, L 109). Where relationships between known life-history strategies and the taxonomical position are known or presumed (e.g. Ocypodidae, Panopeidae, Grapsidae, Sesarmidae), such relationships should of course briefly be discussed. Mainly, however, ecological, reproductive and developmental traits of only the most predominant and better known taxa should be discussed in more detail (where possible, at the species level). This analysis should consider also the quantitative data of larval density in relation to temperature, salinity and chlorophyll concentration. For all rare and lesser known taxa including unidentified larvae, the absence/presence data presented in Table would be largely sufficient as a preliminary set of information. In addition, detailed quantitative data for all taxa should be presented as supplementary material (see comments below).

- I am aware that a more detailed presentation and a more convincing interpretation of the data will require more space in this paper. However, this expansion can be fully compensated by a radical (and necessary!) reduction of the Discussion (almost 16 pp.) and, as a consequence, of the number of references (now 177!). In its present form, this section represents an extended review which, to a great extent, is dealing with larval morphology and taxonomy, rather than an adequate discussion of the data that are actually shown here.

- The authors collected a great set of quantitative data (larval densities, temperatures, salinities, chlorophyll a), not only from 6 sampling stations differing in their distance from the coast, but also from different seasons (7 expeditions in 2 years) and two different sampling depths (horizontal, oblique). In the documentation of the data, however, it seems that these data were often pooled, especially those from surface and subsurface samples, so that much potentially valuable information is lost.

- In the supplementary illustrations (bar charts in Figs. S1 - S8), larval densities are given as mean values without SD and total numbers (n). It remains unclear whether these values were obtained from surface or subsurface samples, or pooled numbers from both.

- For the assessment of different frequencies of successive larval stages (cake charts in Figs. S1 - S8), it is necessary to add total numbers. A fraction of 25%, for instance, could theoretically either correspond to 1 out of 4 individuals, to 150/600, or to 2,000/8,000, which makes a great difference in the meaningfulness of the number "25%".

- The same problem occurs in absence/presence data (e.g. Table 1). Also here, we need to provide total numbers (n) for each taxon. A value of 50% "frequency of occurrence", for example, can correspond to a single larva, if the total n=2; or it might correspond to 500 larvae, if total n=1,000.

- Not being sufficiently familiar with mathmatical models, I cannot evaluate the meaningfulness of patterns in "Expected abundance/probability of occurrence" shown in Figs. 3-5, especially those for entire families. These graphs look nice, but I am not sure if they are sufficiently backed by the available data and the methods used in this study (see below). Some analytical statistics (e.g. ANCOVA) rather than descriptive models might be more appropriate to "predict" or "explain" variations in larval abundance (see Discussion, p. 15).

- Appendix S9 presents insufficient summary data of temperature, salinity and chlorophyll concentrations: Mean values ± SD are given for the 7 sampling dates, but no information on variations among the different stations, nor on differences between the two kinds of sampling. - In the second table, the 6 sampling stations are compared, but no information is given on seasonal or annual variation observed at each distance from the coast. Again, also information obtained from parallel samples taken from the surface and in greater depths is missing.

- In Appendix S10, it is unclear what "higher density" means. Also, this summary table shows only single larval density values (mean, maximum values, n?) obtained at different distances from the coast, but no information on seasonal and annual variation at each sampling station, nor on differences between surface and subsurface samples. Such detailed information should fit in an Appendix table, so that the reader could better evaluate the informative value of the available larval density data and understand the authors’ conclusions presented in the text. If these data are convincing, also the graphs showing "Expected abundance/probability of occurrence" would be better justified.

- Table 1, pp. 10-11: What is, in this context, a "heat map" (different shadings?)? - Overall stage numbers such as "ZI-ZVIII" or "ZI-ZIV, M": This is OK for rare species and those showing no clear tendencies. In some species, however, where sufficient material is available, it would be good also to compare the differential distribution of different larval stages. These could be grouped at least in categories like "early", maybe "intermediate", and "late" (stage numbers then given in parentheses). This would slightly expand the size of this table, but probably enhance its informational value.

Some details of the methods are unclear:

- p. 6, L 140: Were temperature, salinity, and chlorophyll concentrations measured only once per sampling station (only at the surface)? Or are there also records from greater depths (obtained by means of a CTD probe connected to plankton nets that were used to get oblique samples)?

- p. 6, LL 144-146: Why were the samples fractionated in two subsamples, if later "all larvae in the aliquots were identified", i.e. pooled? Or was only one fraction (= 1/2 of each sample) used? In the latter case, it would be weird that half of the information was discarded, although this paper was written 6 years after the end of the sampling programme, providing enough time for complete analyses of the samples. Clarify.

- p. 9, L 214: What is "high depths" (= great depths?), what depths are the authors referring to, and how were such temperature measurements done (see above: CTD probe attached to plankton nets)?

Minor points:

- p. 4, L 88: Something appears to be wrong here. The average discharge of the Amazon is not only"5.7 x 10^2 m^3/year" but >200 m^3/second, or 6,600 KM^3/year (cf. p. 5, L 120). Check.

- p. 5, L 118: Not all sampling intervals were "quarterly" but actually ranging from 2 - 4 months. Clarify.

- p. 4, L 77: Little typo: "harbirs" should be "harbours".

- Caption Fig. 1: after "... 233 km", add: ... "from the NE coast of Marajó Island"; this will clarify that the given distances were not referring to the coast of the mainland.

- We all know it, and after saying it in the title and Introduction of the paper, there’s no need to repeat several times that the ACS is "the largest estuarine / marine system in the world".

6. PLOS authors have the option to publish the peer review history of their article (what does this mean?). If published, this will include your full peer review and any attached files.

Reviewer #1: No

Reviewer #2: No

Reviewer #3: No

---

## [Author Response · Author response to Decision Letter 0]

1 Nov 2021

Reviewer #1: General comments

The manuscript deals with the dispersion of Brachyura larvae in an area of the Amazon continental shelf under the influence of the estuarine plume of the Amazon River. Despite being a relevant topic and there is little information available on the subject, the manuscript presents some points that need to be better presented and explained before the manuscript can be published. My opinion is that the manuscript must be rejected in its present form.

Answer: After reading all the reviewers' comments we considered that many points really needed improvement/clarification. Thus, we are grateful for the reviewer's thorough attention and observations/notes and we hope the changes we made resolve all doubts and lack of clarity in the first version of the text. Since the offshore distribution of crab larvae in the Amazon estuary is very little known and considerable amounts of novel data were obtained in this study, we desire that this paper should eventually be published in an international journal such as Plos One.

 Suggestions for Improvements

Major issues

- The manuscript does not have clear objectives and hypotheses.

- The presented methodology leaves several doubts about the adopted procedures.

Answer: We hope we have corrected these flaws in this second submission of the manuscript.

Pag. 6 line 138. Little information was mentioned about the acquisition of environmental variables, making it impossible to understand how the data were obtained. Data were collected only the surface layer? Other layers of the water column were sampled?

Would the environmental variables obtained in a single layer of the water column be sufficient to understand the complex patterns of larval dispersion on the continental shelf?

Answer: We collected a great set of quantitative data (larval densities, temperatures, salinities, chlorophyll-a), not only from 6 sampling stations differing in their distance from the coast but also from different seasons (7 expeditions in 2 years) and two different sampling depths (horizontal, oblique). We performed several statistical analyses and since we found no significant difference between sampling depths, we decided for pooled, especially those from surface and subsurface samples, and the information was not lost because clear patterns were presented throughout the results, making no sense to separate the samples that do not differ between depths. 

 On the other hand, the environmental variables were not measured in a single layer of the water column; they were measured using a CTD probe (Hydrolab DS 5) that collects data every 0.5 m. We have included paragraph detailing, hoping it became clearer.

Minor issues

Title: “Larval dispersal of Brachyura in the largest estuarine / marine system in the world”

The title of the manuscript does not completely agree with the focus of the manuscript.

Answer: Even though the reviewer argues against the title here, they also comment elsewhere: "The manuscript deals with the dispersion of Brachyura larvae in an area of the Amazon continental shelf under the influence of the estuarine plume of the Amazon River”. From our perspective, this might merely be a grammar issue, which in turn does not necessarily correspond to changes in content. Since the title length already complies with the 250 characters limit, we would rather keep the title as is.

Material and methods

Pag. 4. Line 86. What do the authors consider to be a rare and deep environment?

Answer: Conspicuous is better to describe, thanks for the observation. Rare because finding mesophotic reefs in an area under the influence of the largest river in the world, with high turbidity, is not common (and our samples coincided with this area) and deep referred to having samples along the water column and not just superficial but most of the work recorded with plankton in this region. 

Pag. 6 line 138. The horizontal samplings were carried out during the daytime, at night time, in which tide? Sampling was standardized?

Answer: The collections were standardized and carried out in daytime during the syzygy tide (full or new moon).

Pag. 6 line. 144. What are the criteria used in choosing the fixed collection points? How was depth measured?

Answer: Horizontal and oblique hauls in each site at different coastal distances (23, 53, 83, 158, 198, and 233 km), were chosen because they are representative of regions under the greatest influence of the Amazon estuarine plume (23-83 km from the coast) and also to contemplate the great extension of the Continental Shelf (approximately 300 km) with the distance of 158-233 km with a predominance of oceanic waters.

Pag. 6. Line 142. “V'-shaped, covering up to 75 % of the local depth”. This collection procedure should be better explained.

Answer: Due to the strong undercurrents, the presence of the Great Amazon Reef System (GARS) in some of the places and fluid mud in others, we cannot drag the nets close to the bottom in this region. Thus, we chose to drag up to 75% of the depth of each location, which corresponded to approximately 10, 19, 34, 39, 53 and 80m depths off the coast to the slope. We inserted in the text - the …oblique depths ('V'-shaped, covering up to 75 % of the local depth, approximately 10, 19, 34, 39, 53 and 80m depths off the coast to the slope).

Pag. 7. Line 150. “Simultaneously, water temperature (ºC), salinity, and chlorophyll-a (μg / L) contents were measured, using a CTD probe (Hydrolab DS 5)”. Environmental variables were obtained from the surface layer? No data were obtained in other depths?

Answer: The environmental variables were not measured in a single layer of the water column; they were measured using a CTD probe (Hydrolab DS 5) that collects data every 0.5 m. We have included a paragraph detailing, hope it's clearer now.

Pag. 7 line 158. What keys were used by the authors to identify the different taxonomic levels? How the different stages of larval development were identified?

Answer: As described in the methodology, "All brachyuran larvae in the aliquots were dissected and identified to the lowest taxonomic level possible [1, 45–61] by observing the morphological parameters, such as the arrangement and number of spines and setae in the antennae, antennules, maxilliped, maxilla, abdomen, and telson, under an Axioscope Zeiss A1 optical microscope (Carl Zeiss, Oberkochen, Germany).” The identification keys are those present in the articles indicated: 1, 45-61. And all different stages of larval development were identified. If a certain stage does not appear in the results, it is because it did not occur in the sample. That is, all larval stages were identified if they occurred in the samples, as described in the text.

Results

In general, the analyzes presented do not clearly demonstrate the relationship of environmental factors in the distribution of larvae.

Answer: We stand for our statistical analysis, but we do agree that, perhaps, the contribution of each environmental factor to the final prediction of abundance per taxa was lacking. Thus, to clarify this issue, we included Table S12, where all the numbers there vary from 0 (not important for the predictions at all) to 1 (always important).

Pag. 10, line 237. It is not possible to verify how the estuarine plume affected the different sampling points throughout the sampling period.

Answer: We chose to present the data in such a way as to show the influence of the plume on each taxon, as detailed in Table 1 and Figures 6 and 7. Presenting these data organized by location would add one more table to the manuscript. Is it really necessary? It was a failure to not include figures 6 and 7, as they clearly indicate the plume's boundary. We hope that it became clearer now.

Discussion

Some points mentioned above about distribution models may affect manuscript discussion.

Answer: Although we keep the statistical analyses unchanged because we believe they are the most adequate and even refined in terms of data with a matrix full of zeros and distinct values, which is the case of data on larvae in plankton, the discussion was rewritten following the reviewer’s suggestions and we hope to have clarified better the work.

Reviewer #2

This study summarizes field distributions of Brachyuran crab larvae along the Amazon Continental Shelf. The authors describe larval dispersal of diverse taxa across both spatial (depth/distance offshore) and seasonal scales. To date, relatively few studies have reported extensive field distributions of Brachyura during larval development. Hence, this study provides a valuable contribution to better understanding their dispersal patterns. Please find my comments and suggestions below. Many of my comments are minor suggestions, although there are four major concerns that I hope the authors will address in revision. 

Answer: Thank you very much for your appointments.

First, the phrase “larval dispersion” is used incorrectly and should most likely be replaced with “larval dispersal” throughout the manuscript. In ecology, dispersion indicates investigation of specific distribution patterns, e.g. random, clumped, or uniform, whereas “dispersal” indicates the movement of individuals, e.g. export vs. retention. 

Answer: Ok, “dispersion" was replaced with "dispersal", thanks.

Second, the figures that show model results (Figs. 3 – 5) are missing key information, which make it difficult to confirm the authors’ interpretation of the data. I provided specific comments to this regard below. 

Answer: Ok, we answered below each comment.

Third, the paper should include a summary of model outcomes in the results section to support statements made later in the discussion. For example, the authors state that salinity was a reliable predictor of larval dispersal, while temperature and chlorophyll-a were not; however, statistical support for this claim is lacking. Overall, I find it concerning that an extensive multi-model approach was described in the methods section, but there is little description of the outcomes in the results section (aside from Figs. 3 – 5, which require more detail in their respective captions). 

Answer: Ok, we included the S12 Appendix fill this gap.

Fourth, the impact of the seasonal plume is a central point in the paper. However, statistical/graphical support is lacking (see my comment regarding lines 285 – 297 below). The authors reference Figs. 6 and 7 to support this analysis. However, these figures were not included in this submission. In summary, this study represents a valuable contribution to better understanding the dispersal of larval Brachyurans in coastal systems. However, I recommend that the concerns presented in this review be addressed prior to publication.

Answer: In fact, we forgot to attach the figures, it was our fault. They are attached now and with restructured sentences. We have also included a table of model results. We hope it has become clearer. Thanks for all your concerns. 

Abstract

The importance of the Amazon River seasonal flow and plume is not mentioned until the end of the abstract. Opening the abstract with a sentence or two describing the system would help guide the reader through the results that are summarized just below.

Answer: Ok. We inserted the following sentence: “The Amazon Continental Shelf (ACS) is a complex habitat that receives a large annual freshwater discharge into the ocean, producing a superficial plume and carrying with it large amounts of nutrients to the continental shelf along thousands of kilometers and that sustaining a high biodiversity in the estuary–ocean continuum.”

Lines 31 – 33: This sentence runs on a bit. It would be useful to more clearly highlight the two objectives described: (1) to analyze the composition of larval Brachyuran crabs and (2) to predict the importance of environmental parameters in structuring their occurrence/abundance.

Answer: The sentence was rewritten. Thanks for the suggestion.

Line 33: “A total of 17,759 identified larvae are…” should be “A total of 17,759 identified larvae were…”

Answer: The word “are” was substituted for “were”. Thanks.

Line 41: “(> = 33.5)” should be “(≥ 33.5)”

Answer: It was changed, thanks.

Line 47: Remove “plankton”

Answer: It was excluded.

Introduction

Line 76: Do you mean “harbors”, rather than “habirs” here?

Answer: The correct word is "harbours".

Line 83: Why include “among others”?

Answer: It was excluded.

Lines 101 – 106: I suggest restructuring of this sentence. It might be useful to break it up into (1) parameters of larval composition/distribution and (2) how these relate to environmental profiles.The current organization is a bit convoluted.

Answer: Ok. The text was modified by:

 “In this study, the larval period of Brachyura was evaluated regarding (1) parameters of larval composition/distribution and (2) how these relate to environmental profiles. The first parameters were analised by the larval dispersal extent related to the estuary/plume, and by the frequency of occurrence (FO) in the plume categories. The latter was predicted by the probability of occurrence (PO) and abundance for the temperature, salinity, and chlorophyll-a profiles in the ACS In this study, the larval period of Brachyura was evaluated regarding (1) parameters of larval composition/distribution and (2) how these relate to environmental profiles. The first item was analised by the larval dispersal extent related to the estuary/plume, and by the frequency of occurrence (FO) in the plume categories. The latter was predicted by the probability of occurrence (PO) and abundance for the temperature, salinity, and chlorophyll-a profiles in the ACS with a multimodel approach.” 

Line 108: The “aquatic food chain” is an oversimplification. Consider using “food web” or

“trophic interactions” instead.

Answer: The “aquatic food chain” has been changed to “food web”.

Line 111: I’m not sure what is meant by “…should be distributed on their parental populations…” Are you expecting close proximity to parental populations? Given the common export strategy of estuarine crabs, would this be likely for all Brachyuran crabs in the region?

Answer: Yes, we were expecting close proximity to parental populations, be it from the indentations of the coastal zone or the area of the Great Amazon Reef System (gars). We cannot know for sure at this point, but future studies might use DNA analyses from larvae and from adults in these areas.

Methods

Line 122: “July/2013 to January/2015” should be “July 2013 to January 2015”

Answer: Done. Thanks.

Line 143: “haulsin” should be “hauls in”

Answer: The word “haulsin” was changed to “hauls in”

Lines 169 – 170: The density unit should have not a period after larvae, and the sentence could be restructured for clarity, e.g. “Density (larvae m-3) was estimated by dividing Brachyuran larval abundance by the volume filtered through the plankton net.”

Answer: The sentence was restructured.

Line 176: “…of each species larvae…” should be “…of each larval species…” or “…of each species of larvae”

Answer: The sentence “…of each species larvae” was changed for “…of each larval species”

Lines 214 – 220: Citation(s) to support that this threshold is “widely used”?

Answer: We decided to rephrase that sentence that now reads: 

“It is noteworthy that our multimodel and information-theoretical approach does not allow (and it would be meaningless) to report p-values of any kind [63,68], thus we used the total weight of each variable across all plausible models, varying from 0 (never important) to 1 (always important), as a measure of influence on the predictions. For clarity, we considered a model to be plausible if its Δ_j< 2, which, within the information-theoretical context, means that there is strong empirical evidence that the model is a good approximation to reality (low loss of information) compared to the pool of all models under consideration (see 63;66-67, for instance)”

Results

Line 229 and 233: Replace “amplitude” with “range”

Answer: It was replaced.

Line 230: The parenthetical “begin” should be “beginning” in both occurrences here.

Answer: It was replaced.

Line 238: Remove the “/” between month and year

Answer: It was excluded.

Line 272: Add “and” before Pinnixa

Answer: It was added.

Line 323 – 336: I think that the seasonal plume analysis discussed here is important. However, this section requires visual, model, and/or statistical support. Figure 6 and 7 are referenced in this paragraph but these were not included in the submission. In addition, several of the statements made cannot be supported by Table 1 or Figs. 3 – 5, which do not include seasonal information.

Answer: You are right. The figures have now been included and we have added both N and density detailing in relation to the SS and O drags (S10 and S11 Appendix), so we hope this flaw has been corrected.

Lines 350 – 352: To make this statement, more support is needed in the results section (see previous comment regarding lines 320 – 334).

Answer: With the inclusion of tables S10 and S11 detailing each haul, it is possible to visualize the mentioned statement.

Line 360: Should be “Brachyuran larvae”

Answer: It was corrected.

Lines 363 – 365: Where are the model results that support this statement?

Answer: To clarify the contribution of each covariable for the averaged predictions, we included Table S12. All the numbers in that table vary from 0 (not important for the predictions at all) to 1 (always important).

Line 468: Remove “Anyway”

Answer: It was removed.

Line 496: Is “particular” the best word here?

Answer: No. It was changed by ‘peculiar'.

Line 511: Why change the subtitle structure at this point? All others list the family only.

Answer: You’re right. It was corrected.

Line 539: Same as above – why alter the subtitle structure?

Answer: It was corrected.

Line 541: “larval” before zoea is a bit redundant

Answer: You’re right. It was excluded.

Line 563: Either add a semicolon after “…(Herbst, 1803)” or start a new sentence, i.e. “The

latter is…” The structure of species name reporting also switches to parentheticals here:

“Calappa gallus (Herbst, 1803)” rather than the previous “Calappa sulcata Rathbun, 1898”. For consistency, use the same format throughout.

Answer: A new sentence “The latter is…” has been started. The Calappa gallus species was revised, so the descriptor is in parentheses.

Line 622: “And” should not be capitalized.

Answer: It was corrected.

Table and Figures

Table 1 only shows the frequency of occurrence for one month/year for each group. Based on the supplemental figures, I assume this is the timepoint with the highest density for each individual taxon. If so, indicate this in the table caption. I also suggest characterizing the colors of the heat map described, i.e. frequency of occurrence increases in order of white, light gray, dark gray, and black. Also, “S” is used as an abbreviation for salinity and for sub-superficial sample. Perhaps, it would be clearer to change the latter to “SS”.

Answer: ok, it’s done, thanks!

Fig. 1 caption: It would be useful to identify that the distances listed are kilometers offshore, e.g. “…(23 km, 53 km, 83 km, 158 km, 198 km, and 233 km offshore).”

Answer: Ok, it was included.

Figs. 3 – 5: More information is needed in the caption, which could also be accomplished by adding a legend. For example, what do the different colors represent – different models? Do the shaded regions around each trendline indicate a confidence interval of some sort? Are the y-axis values shown expected abundance per some unit of volume? The methods state that the “final prediction for the expected abundance of each group is given by the product between its predicted abundance and the PO in the reference scenario”. However, the notation in the y-axis label indicates that you are showing a ratio of predicted abundance / PO, rather than the product.

Answer: We have added the proper legends to clarify these issues (please see the updated figures and the legend texts below). The expected abundance was predicted using the median volume and this information was added in the Methods/Data Analysis section. There was a typo in the y-axis label in Figs 3-5, that now reads “Expected abundance*probability of occurrence” (i.e., a multiplication rather than a ratio), as was already correctly stated in the methods.

Fig 3. Final expected abundance (product between the expected abundance and the probability of occurrence) of Brachyura larvae (A: Grapsidae; B: Panopeidae; C: Pinnotheridae) according to the distance from the estuary and net depth in different salinities. The shaded regions refer to the weighted 95% confidence interval. All predictions used the observed median values of temperature (28.4 ºC), chlorophyll-a (7.34 µg / L) and water volume (278.3 m–3).

Fig 4. Final expected abundance (product between the expected abundance and the probability of occurrence) of Brachyura larvae (D: Portunidae; E: Ocypodidae; F: A. rubripes) according to the distance from the estuary and net depth in different salinities. The shaded regions refer to the weighted 95% confidence interval. All predictions used the observed median values of temperature (28.4 ºC), chlorophyll-a (7.34 µg / L) and water volume (278.3 m–3).

Fig 5. Final expected abundance (product between the expected abundance and the probability of occurrence) of Brachyura larvae (G: Calappa; H: Leucosiidae) according to the distance from the estuary and net depth in different salinities. The shaded regions refer to the weighted 95% confidence interval. All predictions used the observed median values of temperature (28.4 ºC), chlorophyll-a (7.34 µg / L) and water volume (278.3 m–3).

Reviewer #3

Review: "Larval dispersal of Brachyura in the largest estuarine / marine system in the world", submitted to PlosOne by F.A. de Lima et al.

This paper describes seasonal variations and regional distribution patterns in the occurrence of brachyuran crab larvae studied in a transect from 23 to 233 km off the NE coast of Marajó Island, Amazon estuary, Brazil. Each transect comprised 6 sampling stations visited from July 2013 to January 2015 during 7 expeditions in approximately quarterly intervals (actually every 2-4 months). Since the offshore distribution of crab larvae in the Amazon estuary is very little known and considerable amounts of novel data were obtained in this study, this paper should eventually be published in an international journal such as Plos One. Before it can be accepted, however, it should undergo a thorough revision and restructuring. Most of my concerns are related to the description of the methods, the presentation of the data, and the detailed discussion of higher taxa (mostly at the family level). While I feel that some aspects are given too much attention, others are neglected or remain unclear (see below). - I suggest that the authors should carefully consider the following points:

- General organization of this paper: It appears to me (especially in the Abstract and in the Discussion section), that the authors put far too much emphasis on the systematic position of the identified larvae rather than to ecological groups and reproductive strategies. For some of the 25 crab taxa identified in this study, the benthic juvenile and adult life-history stages are well known as to their salinity requirements, living and reproducing either in estuarine (i.e. brackish) coastal habitats or in offshore (marine) waters with higher salinities. The data gathered in this field study, especially those of horizontal and vertical distribution of crab larvae (comparing parallel samples from surface and sub-surface water) should therefore be looked at in more detail, mainly in the context of export and retention strategies (for aims of this study, cf. p. 5, L 109). Where relationships between known life-history strategies and the taxonomical position are known or presumed (e.g. Ocypodidae, Panopeidae, Grapsidae, Sesarmidae), such relationships should of course briefly be discussed. Mainly, however, ecological, reproductive and developmental traits of only the most predominant and better known taxa should be discussed in more detail (where possible, at the species level). This analysis should consider also the quantitative data of larval density in relation to temperature, salinity and chlorophyll concentration. For all rare and lesser known taxa including unidentified larvae, the absence/presence data presented in Table would be largely sufficient as a preliminary set of information. In addition, detailed quantitative data for all taxa should be presented as supplementary material (see comments below).

Answer: Thanks. We hope to have filled these gaps in the revised version.

- I am aware that a more detailed presentation and a more convincing interpretation of the data will require more space in this paper. However, this expansion can be fully compensated by a radical (and necessary!) reduction of the Discussion (almost 16 pp.) and, as a consequence, of the number of references (now 177!). In its present form, this section represents an extended review which, to a great extent, is dealing with larval morphology and taxonomy, rather than an adequate discussion of the data that are actually shown here.

Answer: Okay, the Discussion and references have been shortened and the requested detail has been included as there is indeed supporting data.

- The authors collected a great set of quantitative data (larval densities, temperatures, salinities, chlorophyll a), not only from 6 sampling stations differing in their distance from the coast, but also from different seasons (7 expeditions in 2 years) and two different sampling depths (horizontal, oblique). In the documentation of the data, however, it seems that these data were often pooled, especially those from surface and subsurface samples, so that much potentially valuable information is lost.

Answer: Ok, we have detailed the tables to show the abundance in the two different sampling depths.

- In the supplementary illustrations (bar charts in Figs. S1 - S8), larval densities are given as mean values without SD and total numbers (n). It remains unclear whether these values were obtained from surface or subsurface samples, or pooled numbers from both.

Answer: The larval densities are given in S1-S8 as sum of total in each category. The N and the density obtained from surface or subsurface samples are presented now in S10 and S11 Appendix.

- For the assessment of different frequencies of successive larval stages (cake charts in Figs. S1 - S8), it is necessary to add total numbers. A fraction of 25%, for instance, could theoretically either correspond to 1 out of 4 individuals, to 150/600, or to 2,000/8,000, which makes a great difference in the meaningfulness of the number “25%".

Answer: Ok, but the charts/figures become very busy and difficult to see. We resolved this question by detailing the data obtained by hauls, months and distance from the coast in tables.

- The same problem occurs in absence/presence data (e.g. Table 1). Also here, we need to provide total numbers (n) for each taxon. A value of 50% "frequency of occurrence", for example, can correspond to a single larva, if the total n=2; or it might correspond to 500 larvae, if total n=1,000.

Answer: Ok. This was done in tables S10 and S11 in the appendix.

- Not being sufficiently familiar with mathematical models, I cannot evaluate the meaningfulness of patterns in "Expected abundance/probability of occurrence" shown in Figs. 3-5, especially those for entire families. These graphs look nice, but I am not sure if they are sufficiently backed by the available data and the methods used in this study (see below). Some analytical statistics (e.g. ANCOVA) rather than descriptive models might be more appropriate to "predict" or "explain" variations in larval abundance (see Discussion, p. 15).

Answer: There was a typo in the y-axis label in Figs 3-5, that now reads “Expected abundance*probability of occurrence” (i.e., a multiplication rather than a ratio). We have conducted a much more robust statistical analysis than a simple ANCOVA: since our data comprised of counts (number of larvae), many of which were zeros, we used a Hurdle-Poisson approach, which allowed us to separate the expected abundance from the probability of occurrence. In fact, we used many different Hurdle-Poisson models (some of which indeed had an ANCOVA-like predictor) and the results presented were the weighted average (by Akaike weights) predictions given by each of the fitted models. Thus, our models are far from purely descriptive – they rely on a strong, inferential statistical background, avoiding both, unrealistic normality assumptions (as in an ANCOVA model) and p-value based hypothesis testing (which have been subjected to its own scrutiny recently). To clarify the contribution of each covariable for the averaged predictions and to avoid the misinterpretation of a merely descriptive model, we included Table S12 Appendix. All the numbers in that table vary from 0 (not important for the predictions at all) to 1 (always important), which, we believe, is more informative than p-values based on single (“best”) models. Despite the relative simplicity of single models, time and time again, they are not evaluated regarding to goodness-of-fit metrics, residual analysis, and other model validation techniques that are paramount for the correctness and trustworthiness of the p-values. Hence, hypothesis testing using an ANCOVA model in our case would provide straightforward, but completely wrong, results due to the nature of our discrete, zero-altered data. Conversely, we could provide the results of an extended ANODEV based on the “best” Hurdle-Poisson model [similar to what we did in Regional Studies in Marine Science 47 (2021) 101960: https://doi.org/10.1016/j.rsma.2021.101960], but they would be tedious when evaluating several taxa simultaneously. Finally, we tried our best to provide an interesting, computer-intensive data analysis – one for which we stand –, pointing to alternatives to the use of the, often flawed, p-values.

- Appendix S9 presents insufficient summary data of temperature, salinity and chlorophyll concentrations: Mean values ± SD are given for the 7 sampling dates, but no information on variations among the different stations, nor on differences between the two kinds of sampling. 

Ok, but we included Table S12 for clarified the contribution of each environmental factor to the final prediction of abundance per taxa, and cited in the results this sentence: "…for detailed environmental variables in this region, see [27: doi:10.1017/S0025315421000308]”.

 In the second table, the 6 sampling stations are compared, but no information is given on seasonal or annual variation observed at each distance from the coast. Again, also information obtained from parallel samples taken from the surface and in greater depths is missing.

Answer: Ok, it was detailed.

- In Appendix S10, it is unclear what "higher density" means. Also, this summary table shows only single larval density values (mean, maximum values, n?) obtained at different distances from the coast, but no information on seasonal and annual variation at each sampling station, nor on differences between surface and subsurface samples. Such detailed information should fit in an Appendix table, so that the reader could better evaluate the informative value of the available larval density data and understand the authors’ conclusions presented in the text. If these data are convincing, also the graphs showing "Expected abundance/probability of occurrence" would be better justified.

Answer: Ok, it was detailed.

- Table 1, pp. 11-12: What is, in this context, a "heat map" (different shadings?)? - Overall stage numbers such as "ZI-ZVIII" or "ZI-ZIV, M": This is OK for rare species and those showing no clear tendencies. In some species, however, where sufficient material is available, it would be good also to compare the differential distribution of different larval stages. These could be grouped at least in categories like "early", maybe "intermediate", and "late" (stage numbers then given in parentheses). This would slightly expand the size of this table, but probably enhance its informational value.

Answer: This is the only change out of all of the reviewer's excellent suggestions that we do not agree with. Manuscripts that describe in detail which larval stage occur in a particular location are rare, and changing the data that discriminates which stages were found to group them into categories such as "early", "intermediate", and "late" would reduce this valuable information/data. At the same time, it is redundant to group and then put in parentheses the detailed information, which would generate more lines in a table with a lot of detail, as the reviewer himself recognizes. Therefore, we chose to maintain the initial presentation that does change the result at all while being more objective and without artificial categories.

Some details of the methods are unclear:

- Pag. 7, line 150: Were temperature, salinity, and chlorophyll concentrations measured only once per sampling station (only at the surface)? Or are there also records from greater depths (obtained by means of a CTD probe connected to plankton nets that were used to get oblique samples)?

Answer: Abiotic data were obtained in the surface layer and along the water column by a CTD probe (Hydrolab DS 5), which recorded the temperature, salinity and chlorophyll-a at every 0.5 meters of depth. These data in full were used in the mathematical models, without clusters.

- Pag. 7, lines 157-161: Why were the samples fractionated in two subsamples, if later "all larvae in the aliquots were identified", i.e. pooled? Or was only one fraction (= 1/2 of each sample) used? In the latter case, it would be weird that half of the information was discarded, although this paper was written 6 years after the end of the sampling programme, providing enough time for complete analyses of the samples. Clarify.

Answer: Brachyura is one of the most abundant groups in Amazonian zooplankton samples and occurred in high numbers in our samples when compared to other Decapoda. Thus, it was numerically and humanly unfeasible to identify all the individuals of 84 samples with 500mL each, considering that the identification is thorough and done individually, with dissection of the appendices for identification under a microscope. Thus, we chose to subsample all samples in aliquots of 250mL each, which reduced the number of larvae to half without compromising the representativeness of the group. All larvae of the 84 250mL aliquots were identified, totaling 17,759 individuals, and this number was multiplied by the subsampling factor (2) for subsequent calculation of density and elaboration of models.

- Pag. 10, line 231: What is "high depths" (= great depths?), what depths are the authors referring to, and how were such temperature measurements done (see above: CTD probe attached to plankton nets)?

Sorry, the correct is "great depths”. It was a translation problem, thank you.

Minor points:

- Pag. 4, line 87: Something appears to be wrong here. The average discharge of the Amazon is not only"5.7 x 10^2 m^3/year" but >200 m^3/second, or 6,600 KM^3/year (cf. p. 5, L 120). Check.

Answer: You’re right. The correct average is 5.7 x 1012 m³/year. Thanks.

- Pag. 6, line 122: Not all sampling intervals were "quarterly" but actually ranging from 2 - 4 months. Clarify.

Yes, not “quarterly" but 2-4 months. It was changed by: "We sampled in six different locations along the coastal area of Ilha do Marajó to near the slope, from 23 to 233 km away from the coast (Fig. 1), and ranging from 2-4 months from July 2013 to January 2015."

- Pag. 4, line 76: Little typo: "harbirs" should be "harbours".

Answer: You’re right. The correct word is harbours. 

- Caption Fig. 1: after "... 233 km", add: ... "from the NE coast of Marajó Island"; this will clarify that the given distances were not referring to the coast of the mainland.

Answer: Thanks, it was included.

- We all know it, and after saying it in the title and Introduction of the paper, there’s no need to repeat several times that the ACS is "the largest estuarine / marine system in the world".

Answer: Ok, it was excluded.

---

## [Decision Letter · Decision Letter 1]

27 Jan 2022

PONE-D-21-16420R1Larval dispersal of Brachyura in the largest estuarine / marine system in the worldPLOS ONE

Dear Dr. de Lima,

Thank you for submitting your manuscript to PLOS ONE. After careful consideration, we feel that it has merit but does not fully meet PLOS ONE’s publication criteria as it currently stands. Therefore, we invite you to submit a revised version of the manuscript that addresses the points raised during the review process.

We look forward to receiving your revised manuscript.

Kind regards,

Atsushi Fujimura

Academic Editor

PLOS ONE

Journal Requirements:

Additional Editor Comments:

In addition to Comments to the Author below, one of the reviewers provided the following suggestion, and I agree with that.

There is just one (really minor!) point, where I do not agree with a change of phrasing suggested by Rev. #2: "Line 176: “...of each species larvae...” should be “...of each larval species...” or “...of each species of larvae” . - Answer: The sentence “...of each species larvae” was changed for “...of each larval species".

The original phrasing was in fact a little bit awkward in style (actually, only an apostrophe was lacking after species’), but it was semantically unambiguous. In my opinion, a "larval species" (as suggested by the reviewer and adopted by the authors) would be something like an axolotl, i.e. a species that retains a larval form throughout its life. This is not the case in crabs. - Similarly, a "species of larvae" does not exist. There are only larvae that are described and assigned to an already known species (or higher taxon) that had previously been described as an adult form. The description of larvae as supposedly different species occurred typically in the 19th century when complex life histories were largely unknown.

If the Editors and the authors agree with me, I suggest to rephrase p. 8, Line 176 as follows: "The frequency of larval occurrence (FO) was given for each species as a x 100/A, where ...".

Reviewers' comments:

Reviewer's Responses to Questions

**Comments to the Author**

1. If the authors have adequately addressed your comments raised in a previous round of review and you feel that this manuscript is now acceptable for publication, you may indicate that here to bypass the “Comments to the Author” section, enter your conflict of interest statement in the “Confidential to Editor” section, and submit your "Accept" recommendation.

Reviewer #2: (No Response)

Reviewer #3: All comments have been addressed

2. Is the manuscript technically sound, and do the data support the conclusions?

Reviewer #2: Yes

Reviewer #3: Yes

3. Has the statistical analysis been performed appropriately and rigorously? 

Reviewer #2: I Don't Know

Reviewer #3: Yes

4. Have the authors made all data underlying the findings in their manuscript fully available?

Reviewer #2: Yes

Reviewer #3: Yes

5. Is the manuscript presented in an intelligible fashion and written in standard English?

Reviewer #2: Yes

Reviewer #3: Yes

6. Review Comments to the Author

Reviewer #2: The authors have made a significant effort to revise their original manuscript, including satisfactory responses to my major concerns. The revision also benefits from a clearer description of the sampling protocol and a more concise discussion section. Below, I describe minor suggestions for final revision of this manuscript.

Line 62: “Dispersion” is used but I think the authors are commenting on “dispersal” strategies, i.e., larval transport rather than distribution patterns.

Line 103: “analised” should be “analyzed”

Lines 150 – 153: Was the CTD probe attached to the plankton net or were vertical CTD casts done to a particular depth just before or after plankton collection? It is more common to deploy vertical CTD casts just prior to plankton collection, while the ship is stationary. However, the authors state that the CTD “was used to get oblique samples”, implying that the CTD probe was towed.

Line 323: Pinnixa sp. is cited here but this species’ family (Pinnotheridae) is shown in Figure 6B. Why refer to only one species in this family, when the visual evidence provided summarizes the entire family?

Lines 323 – 328: This statement suggests that larvae in families represented by Fig. 6A, 6B, 6C, 6D, and 7H were concentrated in the mid and outer continental shelf during time of greatest discharge (May) and in coastal areas during time of lowest discharge (October). I’m having a hard time seeing this trend in Fig. 7H (Leucosiidae); the greatest density appears to be in the mid continental shelf (158 km) in October.

Lines 328 – 331: I see support for Ocypodidae (Fig. 7A) dispersal from coast to offshore during period of greater plume discharge, but I do not follow the retraction comment. The abundance of this family just appears to be lower overall in October. Couldn’t this just be from seasonal changes in reproduction, rather than reduced plume flushing?

Line 607: Again, “dispersion” or “dispersal”? Discharge more directly affects dispersal, but it is possible that the authors are commenting on distribution patterns (dispersion) of estuarine families.

Table 1 still includes “S” as a haul abbreviation for sub-superficial samples. This is confusing, as “S” is defined as salinity in the table’s caption.

Tables S10 and S11: Table S10 shows distance/location of larvae, while Table S11 shows seasonal changes. Are the larval counts (N and density) combined in these? In other words, are the seasonal densities in Table S11 total or mean values for all distances (Table S10)? This should be clearly defined in the table captions. Also, does “N” represent number of larvae collected? This should also be defined in the table captions.

I suggest adding the Table S12 caption to the supplemental document. Currently, it is only included in the manuscript text.

Reviewer #3: (No Response)

7. PLOS authors have the option to publish the peer review history of their article (what does this mean?). If published, this will include your full peer review and any attached files.

Reviewer #2: No

Reviewer #3: No

---

## [Author Response · Author response to Decision Letter 1]

17 Mar 2022

Editor: General comments

There is just one (really minor!) point, where I do not agree with a change of phrasing suggested by Rev. #2: "Line 176: “...of each species larvae...” should be “...of each larval species...” or “...of each species of larvae”. - Answer: The sentence “...of each species larvae” was changed for “...of each larval species".

The original phrasing was in fact a little bit awkward in style (actually, only an apostrophe was lacking after species’), but it was semantically unambiguous. In my opinion, a "larval species" (as suggested by the reviewer and adopted by the authors) would be something like an axolotl, i.e. a species that retains a larval form throughout its life. This is not the case in crabs. - Similarly, a "species of larvae" does not exist. There are only larvae that are described and assigned to an already known species (or higher taxon) that had previously been described as an adult form. The description of larvae as supposedly different species occurred typically in the 19th century when complex life histories were largely unknown.

If the Editors and the authors agree with me, I suggest to rephrase p. 8, Line 176 as follows: "The frequency of larval occurrence (FO) was given for each species as a x 100/A, where ..."

Answer: Yes, we agree. We agree with the editor's general comments “The original phrasing was, in fact, a little bit awkward in style (actually, only an apostrophe was lacking after species’), but it was semantically unambiguous. The sentence was restructured: "The frequency of larval occurrence (FO %) was given for each species as FO = a × 100/A, where a = number of samples containing the species, and A = the total number of samples." Thanks for your valuable suggestion.

Reviewer #2: General comments

The authors have made a significant effort to revise their original manuscript, including satisfactory responses to my major concerns. The revision also benefits from a clearer description of the sampling protocol and a more concise discussion section. Below, I describe minor suggestions for final revision of this manuscript.

Answer: We are grateful for the reviewer's thorough attention and observations/notes and we hope this paper is ready to be published in Plos One.

 Suggestions for Improvements

Line 62: “Dispersion” is used but I think the authors are commenting on “dispersal” strategies, i.e., larval transport rather than distribution patterns.

Answer: The word “dispersion” was substituted for “dispersal”. Only in the last paragraph of the discussion the word “dispersion" was maintained because it deals with distribution patterns. Thanks for the suggestion.

Line 103: “analised” should be “analyzed”

Answer: The word was corrected. Thanks.

Lines 150 – 153: Was the CTD probe attached to the plankton net or were vertical CTD casts done to a particular depth just before or after plankton collection? It is more common to deploy vertical CTD casts just prior to plankton collection, while the ship is stationary. However, the authors state that the CTD “was used to get oblique samples”, implying that the CTD probe was towed.

Answer: Before plankton collection the CTD deploys vertical casts while the ship is stationary. The text was misspelled and we changed it to: Before plankton samples, the environmental variables water temperature (ºC), salinity, and chlorophyll-a (µg / L) contents were obtained using a CTD probe (Hydrolab DS 5) that was used to get samples every 0.5 meters. Oblique was one of the plankton trawlers, not the CTD.

Line 325: Pinnixa sp. is cited here but this species’ family (Pinnotheridae) is shown in Figure 6B. Why refer to only one species in this family, when the visual evidence provided summarizes the entire family?

Answer: Only Pinnixa sp. was inserted because Austinixa occurs only up to 158km and D. crinitichelis only close to the GARS areas, and does not vary regarding the plume. We cannot generalize this conclusion to D. Crinitichelis because it had a very specific occurrence. 

Lines 325 – 330: This statement suggests that larvae in families represented by Fig. 6A, 6B, 6C, 6D, and 7H were concentrated in the mid and outer continental shelf during time of greatest discharge (May) and in coastal areas during time of lowest discharge (October). I’m having a hard time seeing this trend in Fig. 7H (Leucosiidae); the greatest density appears to be in the mid continental shelf (158 km) in October. 

Answer: The Reviewer is correct. Leucosiidae density is higher on the mid continental shelf (158 km), while only Grapsidae (6A), Panopeidae (6B), Pinnotheridae (6C), and Portunidae (6D) follow the freshwater plume. We restructure as follows:

“During the highest flow period, Pinnixa sp., Grapsidae, Panopeidae and Portunidae larvae were concentrated in the mid and outer continental shelf since the freshwater plume “pushes” the larvae to locations away from the coast. The opposite was observed during the lowest outflow, these larvae concentrated near coastal areas (23 and 53 km from the coast) due to the lower freshwater outflow entering the ACS (Figs 6 and 7, S10 and S11 Tables). 

Lines 331 – 334: I see support for Ocypodidae (Fig. 7A) dispersal from coast to offshore during period of greater plume discharge, but I do not follow the retraction comment. The abundance of this family just appears to be lower overall in October. Couldn’t this just be from seasonal changes in reproduction, rather than reduced plume flushing?

Answer: The Reviewer is correct, we agree. The abundance of this family is greater in the highest flow (May), when there is a reproduction peak. The inclusion of the paragraph: “Furthermore, few taxa were not affected by the Amazon plume. Leucosiidae larval density was higher on the mid continental shelf (158 km), while the density of Ocypodidae is also higher at 158 km in the highest flow (May), decreases in lowest flow period (October).”

Line 609: Again, “dispersion” or “dispersal”? Discharge more directly affects dispersal, but it is possible that the authors are commenting on distribution patterns (dispersion) of estuarine families.

Answer: Reviewer 2 is right. We are commenting on the distribution patterns (dispersion) of estuarine families. Thus, we kept “dispersion” on line 607, and “dispersal” elsewhere.

Tables

Table 1: Still includes “S” as a haul abbreviation for sub-superficial samples. This is confusing, as “S” is defined as salinity in the table’s caption.

Answer: The “SS” abbreviation was kept to identify the sub-surface samples, as it is actually not necessary to abbreviate the word salinity, as it causes confusion with the SS of the Surface Drag. We chose to keep salinity in full and abbreviate only the type of drag. S for salinity was kept only in table S12 as there is no mention of drag in this case.

Tables S10 and S11: Table S10 shows distance/location of larvae, while Table S11 shows seasonal changes. Are the larval counts (N and density) combined in these? In other words, are the seasonal densities in Table S11 total or mean values for all distances (Table S10)? This should be clearly defined in the table captions. Also, does “N” represent number of larvae collected? This should also be defined in the table captions.

Answer: S10 and S11 contain the same larval N, which corresponds to the total number of larvae collected by location (S10) and by month (S11). Thus, for the same group, for example Calappa sp. collected in the SS trawl throughout the study period, the total N is equal to 08 larvae, being (4+2+2) at 83km, 158km and 198km (S10) and (6+2) May and October (S11). In S10, the density corresponds to the sum of the density (and not the average because the data do not have a normal distribution, and the average is not a good estimator) at that location during the entire sampling period. In S11, on the other hand, it corresponds to the sum of all densities of each group in that particular month, regardless of location.

Table S12: I suggest adding the Table S12 caption to the supplemental document. Currently, it is only included in the manuscript text.

Answer: It was added.

We are immensely grateful for the careful revision by the reviewers, which has substantially improved our work.

---

## [Editor Report · Decision Letter 2]

21 Apr 2022

Larval dispersal of Brachyura in one of the largest estuarine / marine systems in the world

PONE-D-21-16420R2

Dear Dr. de Lima,

We’re pleased to inform you that your manuscript has been judged scientifically suitable for publication and will be formally accepted for publication once it meets all outstanding technical requirements.

Kind regards,

Atsushi Fujimura

Academic Editor

PLOS ONE

---

## [Editor Report · Acceptance letter]

6 May 2022

PONE-D-21-16420R2 

Larval dispersal of Brachyura in one of the largest estuarine / marine systems in the world 

Dear Dr. de Lima:

I'm pleased to inform you that your manuscript has been deemed suitable for publication in PLOS ONE. Congratulations! Your manuscript is now with our production department. 

Kind regards, 

on behalf of

Dr. Atsushi Fujimura 

Academic Editor

PLOS ONE